# Title: Exploring the combined use of SMAP and Sentinel-1 data for downscaling soil moisture beyond the 1 km scale

Authors: Rena Meyer[1, 3], Wenmin Zhang[1], Søren Julsgaard Kragh[1, 2], Mie Andreasen[2], Karsten Høgh Jensen[1], Rasmus Fensholt[1], Simon Stisen[2] and Majken C. Looms[1]

Affiliations:
[1]Department of Geosciences and Natural Resource Management, University of Copenhagen, Øster Voldgade 10, 1350 Copenhagen, Denmark
[2]Geological Survey of Denmark and Greenland, Øster Voldgade 10, 1350 Copenhagen, Denmark
[3]Now at: Hydrogeology and Landscape Hydrology Group, Institute for Biology and Environmental Sciences, Carl von Ossietzky University of Oldenburg, Ammerländer Heerstr. 114-118, 26129 Oldenburg, Germany

*Correspondence to*: *Rena Meyer (reme@ign.ku.dk)*

Keywords: soil moisture; downscaling; SMAP; Sentinel-1; CRNS; capacitance probes; k-means clustering; land use cover

Highlights:

- Backscatter produces informative signals even at high resolutions
- At 100-m-scale, the Sentinel-1 VV and VH polarizations are soil moisture dependent
- The downscaling algorithm is improved by introducing land cover dependent clusters
- The downscaled satellite and CRNS soil moisture agree best at the agricultural site

Abstract

Soil moisture estimates at high spatial and temporal resolution are of great value for optimizing water and agricultural management. To fill the gap between local ground observations and coarse spatial resolution remote sensing products we use SMAP and Sentinel-1 data together with a unique dataset of ground-based soil moisture estimates by cosmic ray neutron sensors (CRNS) and capacitance probes to test the possibility of downscaling soil moisture to the sub-kilometre resolution. For a high latitude study area within a highly heterogeneous landscape and diverse land use in Denmark we first show that SMAP soil moisture and Sentinel-1 backscatter time series correlate well with *in situ* observations of CRNS. Sentinel-1 backscatter in VV and VH polarization both show a strong correlation with CRNS soil moisture at higher spatial resolutions (20 m - 400 m) and exhibit distinct and meaningful signals at different land cover types. Satisfying statistical correlations with CRNS soil moisture time series and capacitance probes are obtained using the SMAP Sentinel-1 downscaling algorithm. Accounting for different land use in the downscaling algorithm additionally improved the spatial distribution. However, the investigated downscaling algorithm does not fully account for the vegetation dependency at sub-kilometre resolution. The study suggests that future research focussing on further modifying the downscaling algorithm could improve representative soil moisture pattern at fine scale since backscatter signals are clearly informative.

## 1. Introduction

Soil moisture, the amount of water in the vadose zone, is an important state variable in the hydrological cycle controlling water exchange between land-surface and atmosphere. It influences rainfall-runoff processes and the water availability for plants (Ochsner et al., 2013). Hence, it has relevance for water management, agricultural optimization (Vereecken et al., 2008) and natural hazards such as drought, floods (Grillakis et al., 2016) or forest fires (Bartsch et al., 2009; Chaparro et al., 2016).

Different applications require information about soil moisture at different spatio-temporal scales from small size centimetres-scale, to e.g. monitor growth of a single plant, to hundreds of kilometres for climate forecasting (Sabaghy et al., 2018). For application in water management and agricultural optimization in mesoscale catchments (sizes between 100 and 10000 km$^2$), soil moisture information at a spatio-temporal resolution of few hundred meters and close to daily time intervals is desirable (Sabaghy et al., 2018). Complementary to other conventional observation (i.e., groundwater head and discharge, Meyer et al., 2018a) and non-conventional observation data (e.g., groundwater ages; Meyer et al., 2018b, 2019), soil moisture can serve for calibration and validation of hydrological models (Shahrban et al., 2018) and hence contribute to enhancing their prediction accuracy.

Existing techniques to measure spatio-temporal dynamics of soil moisture at field scale (reviewed by Vereecken et al., 2014) include sensor networks with a footprint of few cubic-centimetres, hydro-geophysical techniques, cosmic ray neutron sensors (CRNS) with footprint of a few hundred meters (Andreasen et al., 2016) and (satellite) remote sensing with a footprint of tens of kilometres (Mohanty et al., 2017). Variations in soil texture, topography, climate, land use/cover (LUC) result in a high spatial variability of soil moisture even on field scale (Peng et al., 2017; Vereecken et al., 2014). These variations are most prominent in absolute soil moisture values whereas the temporal dynamics are less influenced (Peng et al., 2017). While ground observations provide a high temporal resolution of soil moisture dynamics, they are less efficient in capturing spatial variation on catchment/field scale due to their small footprint.

Soil moisture observation from space, on the other hand, provide large spatial coverage but is limited to a coarse spatial resolution. Within the last fifteen years NASA and ESA launched satellite missions dedicated for soil moisture observations, i.e. SMAP (in 2015) and SMOS (in 2009), respectively. These missions provide freely available near surface soil moisture observations several times a week over the whole globe with a high sensing accuracy of soil moisture but a relatively coarse resolution of 36-40 km (Mohanty et al., 2017). The satellites carry passive L-band radiometers with a frequency of 1.4 GHz that receive natural radiation in form of brightness temperature emitted from the earths' surface (Das et al., 2014). Microwaves at this frequency are sensitive to the dielectric properties of the upper few centimetres of the soil. The dielectric constant varies with the water content with values between 3 [-] (dry conditions) and 80 [-] (fluid water) which make the radiometer suitable for soil moisture monitoring.

To overcome the limitations in spatial resolution of the radiometer derived soil moisture data, different downscaling approaches have been investigated. Many different techniques exist that either use a statistical correlation (e.g. Mascaro et al., 2011, 2010) or physically based models to combine satellite and land surface characteristics (e.g. Fang et al., 2018), satellite and ground observation (e.g. Ridler et al., 2014), satellite derived indices (e.g. Fang et al., 2013; Peng et al., 2016, Tagesson et al., 2018) or different satellite derived products (e.g. thermal, optical or active-passive microwave data) (e.g. Entekhabi et al., 2014; Wagner et al., 2007) with the aim to downscale the relatively coarse scale soil moisture observations from passive radiometers, e.g. SMAP and SMOS (Tagesson et al., 2018). Two recent review papers by Peng et al. (2017) and Sabaghy et al. (2018) give comprehensive summaries of different downscaling approaches including a categorization and their respective advantages and disadvantages.

Initially, the SMAP mission was composed of a passive L-band radiometer and an active L-band SAR (synthetic aperture radar) to provide optimal data, e.g. coincidence in revisit time, incident angles and frequencies for using an active-passive downscaling approach. However, the active radar failed after a few months of operation and therefore the use of other SAR satellites have been explored. The ESA mission of Sentinel-1 SAR satellites seems particularly suitable for this approach because of the high resolution of up to 10 m x10 m and a revisit time of 3-5 days. However, uncertainties arise from the mismatch of revisit times between SMAP and Sentinel-1 and changing incidence angles of Sentinel-1 that needs to be corrected for. These effects might only have a limited influence on accuracy (He et al., 2018). Moreover, Sentinel-1 carries a C-band SAR (at frequency of 5.405GHz) which is more influenced by vegetation and surface roughness than the L-band SMAP radiometer (Calvet et al., 2011). Nevertheless, He et al. (2018) compared different SMAP Sentinel-1 downscaling algorithms at spatial resolutions of 9 km, 3 km and 1 km and evaluated the soil-moisture-based downscaling algorithm as highly accurate (particularly at the coarser resolutions). Global soil moisture products exist, e.g. provided by NASA at resolutions of 3 km and 1 km, that are derived by the combined SMAP Sentinel-1 downscaling approach applied to downscale

brightness temperature and afterwards transforming it to soil moisture or directly to downscale soil moisture (Das et al., 2019). The high resolution NASA soil moisture product covers the globe from latitudes +60 (North) to -60 (South) (e.g. SMAP/Sentinel-1 L2 Radiometer/Radar 30-Second Scene 3 km EASE-Grid Soil Moisture). However the temporal and spatial coverage at our investigation area at a latitude of +56 (North) is incomplete for the 1 km product. Following the satisfying results of the downscaling approach at the medium resolution (1 km, 3 km and 9 km) by He et al. (2018) and Das et al. (2019), we investigate the applicability of the same approach on high resolution (sub-kilometres scale).

The applicability of remotely sensed soil moisture estimation is strongly related to its accuracy (Colliander et al., 2017). Therefore, ground-based soil moisture measurements, mostly from soil moisture probes, are used to validate remotely sensed soil moisture products. One major challenge of the validation task is the mismatch of spatial scales between point measurements on the ground and large-scale satellite products (Bircher et al., 2012a; Colliander et al., 2017). Often ground measurements are upscaled to the satellite resolution, introducing uncertainties as the absolute values of the small scale sensors are not representative (Colliander et al., 2017). Innovative ground-based soil moisture measurements with the cosmic ray method provide soil moisture information over a footprint of a few hundred meters (Andreasen et al., 2016) and have recently been used for validating satellite derived soil moisture on original sensing scale (e.g. Montzka et al., 2017; Ochsner et al., 2013). However, it has not yet been applied in the context of validation of downscaled soil moisture products. The expected advantage of using CRNS for validation of downscaled soil moisture at a few hundred meter scale is the similarity in scale and hence a better comparability of the two observation methods.

In the present study we first focus on an in-depth analysis of soil moisture from ground observations and remote sensing data from SMAP and Sentinel-1. We present a comparison of the ground-based soil moisture measurements at three CRNS sites (Andreasen et al., 2016) and a dense network of 30 capacitance probes (Bircher et al., 2012b) with SMAP derived soil moisture and Sentinel-1 co- and cross-polarization backscatter. Afterwards we investigate the feasibility to apply the SMAP Sentinel-1 downscaling approach of soil moisture to estimate spatially distributed soil moisture at a sub-kilometre resolution and validate the downscaled soil moisture with the *in situ* soil moisture estimates by CRNS and capacitance probes. Moreover a modification of the downscaling algorithm is proposed in which we account for the vegetation dependency of the algorithm parameters using a k-means cluster analysis. We choose the Ahlergaarde catchment in Western Denmark as study area since it has been subject to many hydrological studies within the HOBE projects (Jensen and Refsgaard, 2018) and hence provides a rich data set of high quality observation data. Furthermore, it is a catchment of the national water resources model (DK-model, Henriksen et al. 2003) and soil moisture at the hundred meter scale is of high interest for potential application in hydrological modelling. Reasons for downscaling soil moisture in this area include (1) the variability of soil texture, LUC and the relatively small size of agricultural fields and (2) the incompleteness of existing soil moisture data from NASA at a spatial resolution of 1 km and below.

## 2. Methods
### 2.1 Study area

The Ahlergaarde catchment, a sub-catchment of the Skjern catchment, is located in western Denmark, at about 56° latitude, and covers an area of about 1058 km² (Fig. 1). The maritime climate with mild winters and cool summers is dominated by a westerly wind regime with frequent rain. The mean annual precipitation is about 990 mm, maximum in autumn, minimum in spring. Mean annual evapotranspiration and mean temperature are 575 mm and 8.2°C, respectively (Jensen and Illangasekare, 2011). The topography is relatively flat (up to 125 m in the East and at sea level in the West) (Jensen and Illangasekare, 2011). The surface geology in the area is characterized by glacial outwash plains consisting of Quaternary sand and gravel with some moraine till. The texture of the topsoil varies across the area and is dominated by fine to coarse sand from glacio- fluvial and glacio- lacustrine origin (Jensen and Refsgaard, 2018). The Skjern catchment is characterized by agricultural use for crop (55%) and pasture (grass, 30%), followed by forest (7%), heathland (5%) and urban areas (2%) (Jensen and Illangasekare, 2011). The average sizes of agricultural fields in the area are less than 100 ha (Stelljes et al., 2017). The main growing seasons are spring and summer with harvesting in late summer and autumn. Due to the predominantly high permeable sandy soils with low water retention capacities, groundwater is abstracted for irrigation in approximately 50% of the catchment area during the summer months from May to August with an average annual demand of 20 mm/year and up to 55 mm/year in dry years (e.g. 2014; Jensen and Refsgaard, 2018) which correspond to two to five times the demand of domestic and industrial water (Jensen and Illangasekare, 2011; Jensen and Refsgaard, 2018).

In 2007, a long-term hydrological observatory, HOBE, was set up in the Skjern catchment with the aim to enhance the understanding of hydrological processes at catchment scale and the impacts of anthropogenic and natural

changes such as LUC and climate change (Jensen and Illangasekare, 2011). In the course of these multi-disciplinary investigations, soil moisture monitoring has been implemented with a network of capacitance probes (Bircher et al., 2012) as well as three CRNS systems for stationary measurement of temporal dynamics of soil moisture at the dominant LUC types, agriculture, heathland and forest (Andreasen et al., 2020) (Fig. 1). Daily precipitation data is available at the agricultural field site (Voulund) and forest field site (Gludsted) (DMI.dk, 2021, Fig. 2, a). In the present study soil moisture from these ground observations and from remote sensing are used in the period from January 1$^{st}$, 2017 till May 31$^{st}$, 2019. This period includes exceptionally dry periods in summer 2018 and winter 2018/2019 (Fig. 2, a).

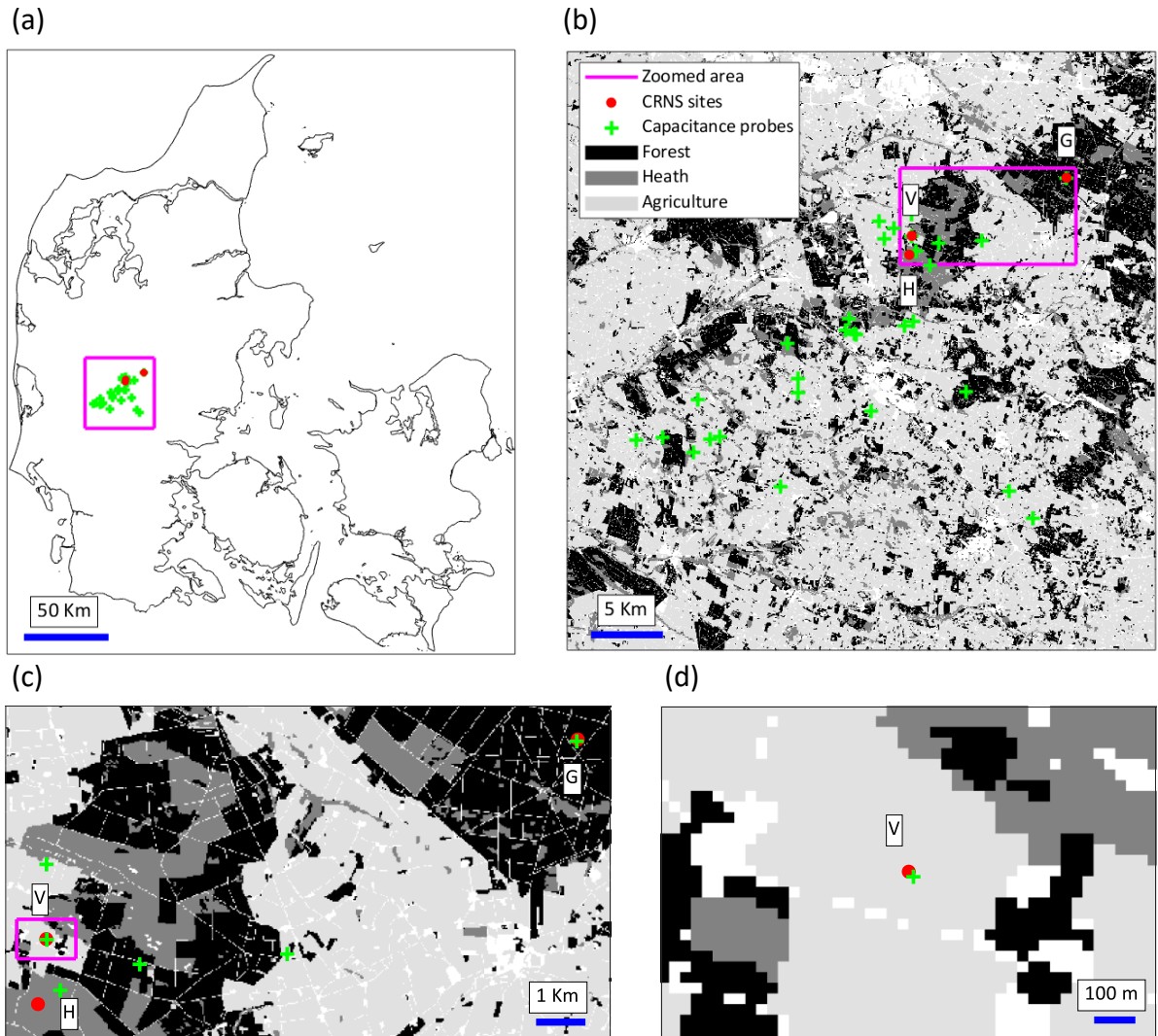

**Figure 1:(a) Location of the study area in Western Denmark with the SMAP pixel extent. Closer look into the area on a LUC map: (b) showing the entire study area, (c) a zoom encompassing all three CRNS stations, for which the results are presented and (d) a close look on one of the specific field sites, Voulund (agricultural field). Ground observations are indicated with green crosses = location of capacitance probes and red dots = CRNS stations at the field sites V=Voulund, H=Harrild and G=Gludsted.**

**2.2 The downscaling algorithm**

In the present study, the active-passive downscaling approach based on Das et al. (2011), Das et al. (2014) and He et al. (2018) is applied. Hereby, the passive radiometer soil moisture data of high accuracy but low spatial resolution is combined with high resolution active SAR. The SAR actively emits electromagnetic pulses and

measures the transmitted energy from the earth surface in form of backscatter (e.g. Das et al., 2014). The downscaling principle relies on the negative relation between brightness temperature and co-polarized SAR backscatter over the dielectric constant which is positively related to soil moisture.

The algorithm is adapted from He et al. (2018) who modified the baseline SMAP Sentinel-1 algorithm of Das et al. (2011, 2014, 2018) by directly downscaling the soil moisture product to spatial resolutions of 9 km, 3 km and 1 km instead of downscaling brightness temperature and subsequently transforming it to soil moisture (He et al. 2018). One major advantage of this technique is that no additional data, such as ground temperature or vegetation water content is needed, which are often difficult to obtain at the relevant resolution, but necessary to invert soil moisture from brightness temperature (He et al., 2018). The downscaled soil moisture at fine scale $\theta_{fine}$ is estimated by

$$\theta_{fine} = \theta_{coarse} + \beta\left[\left(VV_{fine} - VV_{mean\_coarse}\right) + \Gamma\left(VH_{mean\_coarse} - VH_{fine}\right)\right]$$

Here, $\theta_{coarse}$ is the soil moisture retrieved from SMAP, VV and VH are the co-polarization and cross-polarization backscatter from Sentinel-1, respectively. *Fine* means at target resolution and *coarse* at SMAP resolution. Moreover, the downscaling algorithm involves the estimation of two parameters, $\beta$ and $\Gamma$. For clarification, we use $\sigma_{VV}$ and $\sigma_{VH}$ to refer to the original SAR backscatter in co-polarization and cross-polarization and VV and VH for the converted backscatter to dB, respectively. They relate as follows:

$$VV\,[dB] = 10 \cdot log10(\sigma_{VV})$$
$$VH\,[dB] = 10 \cdot log10(\sigma_{VH})$$

$\beta$ relates to the sensitivity of soil moisture to co-polarization radar backscatter ($\sigma_{VV}$) and can be estimated as the slope of a linear regression of $\frac{\theta_{SMAP}}{VV_{coarse}}$ time series. $\beta$ is generally assumed to be invariant in time and space. $\Gamma$ represents the sensitivity of temporal changes in co-polarization to cross-polarization and is calculated based on differential pairs of co- and cross-polarization, $\Gamma = \frac{\delta VV}{\delta VH}$ , at the fine scale (Das et al., 2018; He et al., 2018). The term $\Gamma * \left(VH_{mean\_coarse} - VH_{fine}\right)$ describes the scale heterogeneity of vegetation and surface roughness as the cross-polarization backscatter deviation from fine to coarse scale. This needs to be removed from the fine-scale correction of VV to obtain local soil moisture changes.

## 2.3 Data used:

### 2.3.1 SMAP (L-band)
The SMAP satellite mission by NASA was launched January 25[th], 2015 and provides soil moisture measurement with an accuracy of 4% (at the original 36 km resolution of the SMAP) in the top 5 cm of the soil at different lateral scales, down to 3 km, obtained from a combination of a passive L-band radiometer and an active L-band SAR (Entekhabi et al., 2014), both operating with a constant incidence angle of 40°. The passive radiometer is more sensitive to water in the soil, providing measurements at ~40 km resolution, while the radar is more influenced by surface roughness and vegetation structure and can provide much higher resolutions. As the active SAR radar failed in July 2015, other SAR satellites have been investigated to be used in combination with the SMAP radiometer.

In this study the SMAP radiometer enhanced Level-3 soil moisture product (L3_SM_AP), provided as a daily composite on a 9 km EASE-grid were obtained (O'Neill et al., 2018) for January 2017 to 31[st] May 2019. The daily mean of the ascending and descending product was used, unless only one of them was available. The Ahlergaarde catchment is covered by 21 SMAP 9 km EASE grid pixels.

### 2.3.2 Sentinel-1 (C-band):
The Sentinel-1 satellite mission of ESA's Copernicus programme comprises two polar-orbiting active radar satellites that were launched in 2014 and 2016, respectively. At high latitudes Sentinel-1 has a revisit frequency of about 3-5 days. The C-band SAR of Sentinel-1 provides high resolution, weather independent microwave backscattering in dual polarization ($\sigma_{HH}+ \sigma_{HV}$ and $\sigma_{VV}+ \sigma_{VH}$ ). It operates with an incidence angle range of 20° to 46°. In order to use the two satellites in combination, incidence angles of Sentinel-1 need to be corrected to a reference angle. Here we use a reference angle of 40°, corresponding to the SMAP incidence angle. The radar backscatter, $\sigma_{\theta_i}^0$, obtained at an incidence angle, $\theta_i$, was normalized to a reference angle, $\theta_{ref}$, after the method of Mladenova et al. (2013) which was also applied by He et al. (2018):

$$\sigma_{ref}^0 = \frac{\sigma_{\theta_i}^0\, cos^n\,(\theta_{ref})}{cos^n\,(\theta_i)}$$

with n=2 (Mladenova et al., 2013). In principle, the exponent n is roughness dependent and varies between 0.2 and 3.4 (Mladenova et al., 2013). He et al. (2018) evaluated a value of n=2 as suitable for a similar application as in this study. Based on a comparison of VV and VH time series with and without the incidence angle correction, we concluded that an exponent of 2 is feasible because it improves the time series by reducing the noise while still showing a dynamic behavior. The effect of the incidence angle correction was higher on VV than on VH. In this study, Sentinel-1 A and B Level -1 GRD data in Interferometric Wide swath mode (IW) with a full resolution of 10 m x 10 m was acquired from Google Earth Engine and pre-processed with thermal noise removal, radiometric calibration and terrain correction (Filipponi, 2019). 428 Sentinel-1 images cover the Ahlergaarde catchment during the study period from January 1$^{st}$, 2017 till May 31$^{st}$, 2019. In order to investigate the trade-off level between noise and signal, temporal dynamics of VV and VH backscatter at different aggregated scales are explored at three field sites. Sentinel-1 VV polarization relates to a combination of soil moisture, biomass and vegetation structure while Sentinel-1 VH polarization is assumed to be mostly sensitive to biomass and vegetation structure. For a deeper investigation of the spatial pattern information content of the Sentinel-1 data, an unsupervised data driven k-means cluster analysis (Lloyd, 1982) is performed based on four parameters, the temporal mean and the standard deviationof both the VV and the VH backscatter at target resolution within the study area. The clustering is performed for different spatial aggregation levels (results will be shown for 20 m, 100 m and 1000 m) and different number of cluster groups (2-6). The standard deviation within each cluster does not decrease when applying more than three clusters, hence the data contains information to differentiate clearly between three clusters, but not necessarily more.

Two cross-ratios of the polarized backscatter are calculated to further investigate the relation to biomass and soil moisture:

1) VV/VH is similar to Γ and calculated based on dB converted backscatter as VV/VH=10*log10 ($\sigma_{VV}$)/10*log10 ($\sigma_{VH}$) [dB/dB].

2) VH/VV=10*log10 ($\sigma_{VH}/\sigma_{VV}$) [dB] is first calculated in original backscatter and afterwards converted to dB. Harfenmeister et al. (2019), Veloso et al. (2017) and Vreugdenhil et al. (2018) found that the ratio VH/VV better relates Sentinel-1 backscatter to biomass and compensate for effects of soil moisture and radiometric instability of sensors.

### 2.3.3 Data processing:

For the study period 881 SMAP and 428 Sentinel-1 images were available. The first two years of the Sentinel-1 mission were excluded because the data was not as regular and did not show seasonal variation as the images for the proceeding years used in this study. There were 24 SMAP images without data which were removed. Soil moisture estimates from satellite are erroneous when the ground is frozen (dielectric constant of frozen water is 2 - 3 [-]). Therefore, 77 SMAP data points were removed when the air or soil temperature, measured at the field site Voulund, were low in the winter months and the soil moisture estimates thus were unrealistically low. Further, to ensure optimal downscaling results the satellite data was reduced to only those days where both SMAP and Sentinel-1 data is available. This synchronization reduced the data set to 377 images. with a resulting average frequency of two to three days. Sentinel-1 pixels that show values relating to buildings (identified by high VV) and lakes (identified by low VV) were removed from the Sentinel-1 data set. In details this means: all VV and VH backscatter that were positive were replaced by NaN. Artefacts in the VV and VH data were removed when backscatter values were lower than -40. Lakes and open water were removed when mean VV or mean VH backscatter were lower than -19. Pixels that had more than 30 NaN out of the 377 images were removed for the whole period. Finally, to smooth the Sentinel-1 data for all analysis a temporal moving average of five images was applied to all remaining data at the resolution investigated.

### 2.3.4 Ground observation – validation data

*In situ* ground observations of soil moisture are available from installed capacitance probes and stationary CRNS. The data and set up are briefly presented here while details can be found in the respective publications. In the course of the HOBE project, a network of 30 observation stations (Fig. 1; green crosses) each equipped with five capacitance probes (Decagon 5TE sensors) to monitor soil moisture at different depths (at 0-5 cm, 20-25 cm and 50-55 cm, and in the litter layer where applicable) have been installed in the Ahlergaarde catchment in 2009 (Bircher et al., 2012b). The locations were chosen with respect to representing the variability in LUC and soil texture (Bircher et al., 2012a). Hence, 22 probes are located in agricultural land, 4 in heathland and 4 in forest. The sensors record soil moisture each 30 min and are representative for a soil volume of about 300 cm$^3$ (Decagon Devices Inc, 2016). A recent study investigated the representativeness of these capacitance probes and concluded that dynamics in soil moisture exhibit low deviation while absolute values can be difficult to obtain precisely (Denager et al., 2020).

Within the framework of HOBE, three stationary CRNS stations were established in 2013 and 2014 (Andreasen et al., 2020) (Fig. 1) and data collection is still going on. The field sites are located within 10 km of each other and

represent the three main LUC of the catchment: agricultural land (Voulund), heathland (Harrild) and forest (Gludsted). Hourly CRNS intensity is measured using the CR1000/B system of Hydroinnova LLC, Albuquerque, New Mexico. Following Poissonian statistics, the relative measurement uncertainty of neutron intensity, N, decreases with increasing neutron intensity and the standard deviation equals $N^{0.5}$. The measured CRNS intensity is corrected for variations in barometric pressure, atmospheric water vapour and incoming cosmic-ray intensity using data from the neutron monitor data base (nmdb.eu). The measured CRNS intensity is sensitive to soil moisture in the upper decimeters of the ground within an area of hectometers (Andreasen et al., 2017). The standard $N_0$-calibration function (Desilets et al., 2010) was used to convert CRNS intensity measurements to volumetric soil moisture (Andreasen et al., 2020). Daily moving averages were calculated to obtain acceptable statistics.The influence of the vegetation water content on the CRNS estimated soil moisture is low at the study site because of the limited change in vegetation cover in the heathland and pine forest and the relatively low amount of biomass in the agriculture (8 t/ha consisting of ~15% cellulose and 85% water, Andreasen et al., 2020). The impact of the vegetation cover on the CRNS intensity using field measurements of neutrons at two energy ranges and neutron transport modeling (Monte Carlo N-Particle code version 6, MCNP6) showed very little impact of the vegetation cover at the agricultural site compared to bare soil conditions (Andreasen et al., 2020, Figure 4).

 The footprint of the CRNS varies slightly in space and time. However, the sensor sensitivity is highest in the close vicinity of the probe and decreases exponentially with distance from the sensor. The location of the sensors has been carefully chosen by placing them in same soil type and far enough from the next LUC type to prevent influence/mixture of different LUC signals. Furthermore, Ahlergaarde catchment is situated on a glacial outwash plain, and the study area is characterized by homogeneous soil (sandy and stratified soil with similar chemical composition). Therefore, changes in the vertical and horizontal footprint area are not expected to affect the CRNS signal significantly. A network of capacitance probes (please note that this network is not the same as used in our manuscript, but specifically set up to validate/compare the CRNS estimates), are placed strategically in the vicinity of the CRNS. The long time series of CRNS estimated soil moisture has been shown to be very robust in comparison to the average of these measurements (Andreasen et al., 2020) . Finally, The same data set has been successfully used to improve the closing of the water balance by Denager et al. (2020).

One challenge in the validation of satellite derived soil moisture with ground observations is the difference in scales and representation of vertical sensing depth intervals. To test whether it is reasonable to compare daily mean soil moisture estimates of the remote sensing product with ground observations from different methods representing soil moisture at different depths intervals, the relation of near surface soil moisture (0-5 cm) and soil moisture at 20-25 cm depth measured by capacitance probes is investigated by a linear regression analysis (Supplemental Material; Fig. S1). An acceptable correlation at the majority of the stations support that the comparison is reasonable.

**2.3.5 Spatio-temporal comparison of remotely and in situ soil moisture estimates**
To allow a better comparison of the estimated spatial soil moisture a subset of 16 capacitance probe locations (three in heathland and 13 in the agriculture) was chosen that meet the criteria that probes at 2.5 and 22.5 cm depth show a good correlation in temporal dynamics and cover both the dry summer in 2018 and the wet winter of 2017/2018 (Supplemental Material Fig. S1). The relatively low data quality, incomplete time series and shift in absolute values at the agriculture stations relate to the disturbances caused by the removal and followed reinstallation several times a year to allow ploughing of the fields.
For each of the 16 station the soil moisture range was computed and compared to the same analysis using the downscaled results. To make the analysis more robust the stations were first ordered according to their absolute value and assigned a rank (R). The mean summed difference (mSDR) of the observed rank ($R_{obs}$) and downscaled rank ($R_{dow}$) was calculated,

$$mSDR = \frac{\sum_{i=1}^{N=16} (R_{obs,i} - R_{dow,i})}{N}$$

and used to evaluate how well the downscaled soil moisture agrees with the *in situ* estimates by capacitance probes. The smaller the SDR the better the data agree. For the 16 stations the optimal mSDR is 0 [-], i.e. observed and downscaled soil moisture values are ranked identically. For completely random data, with no correlation between observed and downscaled soil moisture ranks, the average mSDR is 5.3 [-], while the maximum mSDR is 8 [-]. These values were estimated using 50 million random combinations of the 16 soil moisture samples.

**3.  Results**
**3.1 Comparison of remote sensing data with *in situ* measurements of daily soil moisture**

### 3.1.1 SMAP

The temporal dynamics of the soil moisture retrieval from the 21 SMAP pixels are very similar and show strong seasonal dynamics. The SMAP derived mean soil moisture content is high in autumn and winter and low in the summer (Fig. 2, b). High variability of the 21 SMAP pixels coincide with precipitation events and high soil moisture content (Fig. 2, b & f). Overall, the soil moisture estimates by SMAP mimic the trends seen in CRNS (Fig. 2, c) and capacitance probes (Fig. 2, d), showing a significantly drier summer period in 2018 compared to a wetter summer in 2017. Also the winter 2018/2019 is significantly drier compared to the winter 2017/2018. The spring to autumn period in 2018 (Fig. 2, e-h) shows that SMAP soil moisture (Fig. 2, f) increases abruptly in response to rain events (Fig. 2, e) while the ground soil moisture products of CRNS (Fig. 2, g) and capacitance probes (Fig. 2, h) respond more gradually. The difference in response is most likely a result of the vertical sensing depth of the different methods, i.e. SMAP <5 cm, CRNS approximately 0-25 cm and capacitance probes 0-5 cm.

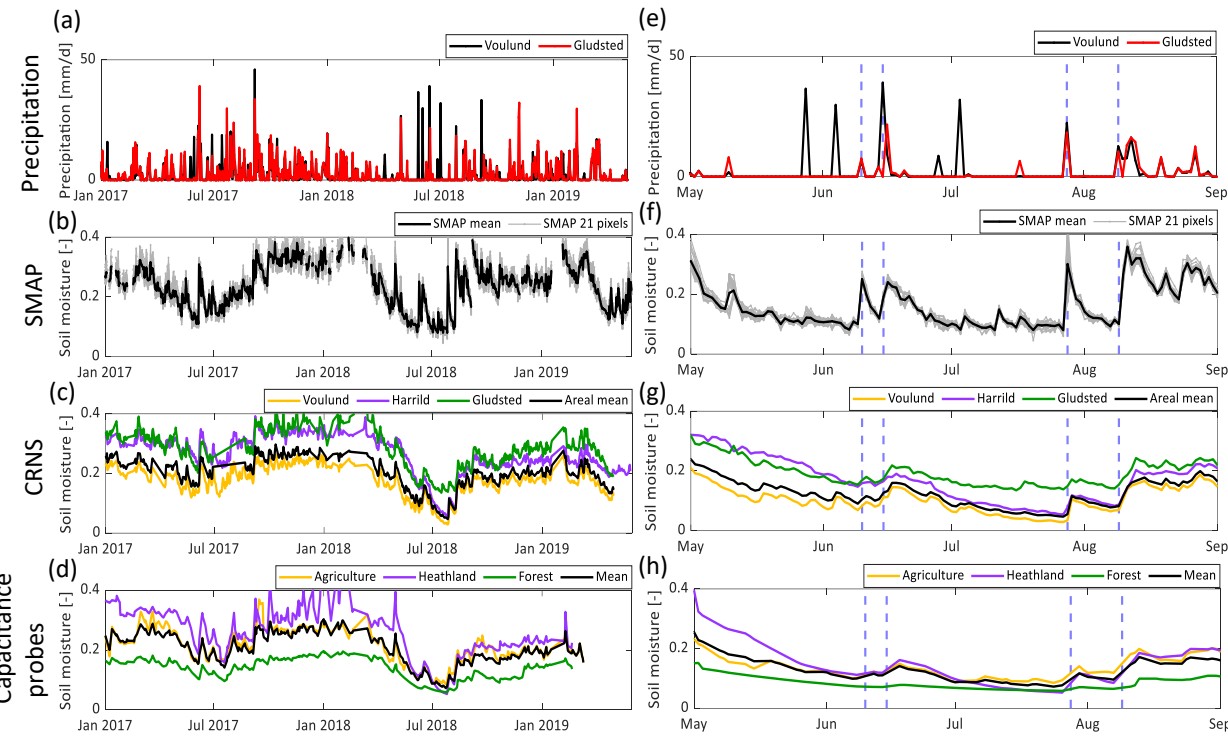

**Figure 2: (a) Precipitation at two field sites, Voulund (agriculture) and Gludsted (forest). Differences in peaks relate to irrigation at the agriculture site (Source of precipitation data: DMI.dk, 2021, open data). (b) Time series of daily soil moisture estimates by SMAP for all 21 pixels and averaged over the whole area. (c) Time series of CRNS derived daily soil moisture at the three field sites Voulund (agriculture), Harrild (heathland) and Gludsted (forest). (d) Time series of daily soil moisture** (in adepth of 0-5cm) **derived by capacitance probes and representing the mean of the three LUC. The left panel (a-d) shows the time series for the entire 2.5 year study period while the right panel (e-h) shows the growing period May-September in 2018.**

The linear regression of soil moisture estimates derived from SMAP and CRNS at the three different field sites show a good correlation between the different sensing methods with an acceptable areal mean RMSE (0.056 [-]) and a high correlation ($R^2$=0.7) ( Fig. 3) even before any downscaling attempt and hence represents a big difference in sensing scales. The same holds for the correlation of soil moisture estimates derived from SMAP and capacitance probes at 0-5 cm depth. The mean of the 30 capacitance probes fits very well the mean of the SMAP estimated soil moisture (Fig. 3), as also observed previously with SMOS data (Bircher et al., 2012).

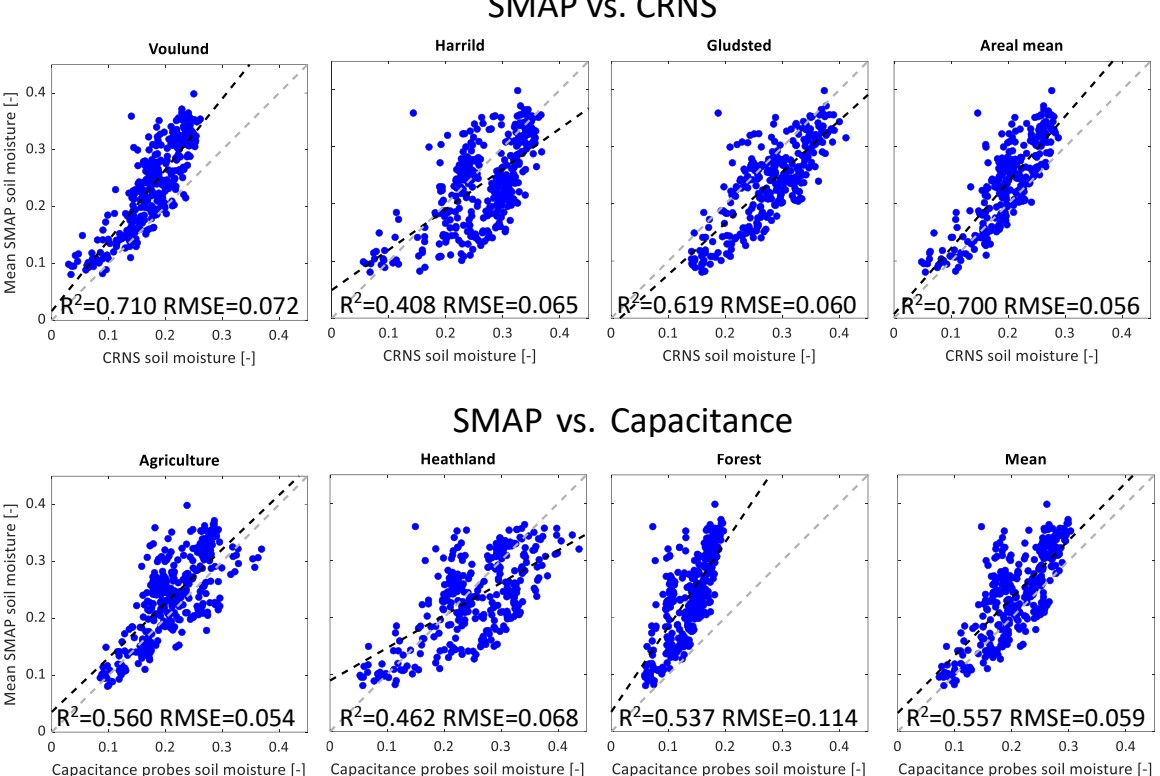

**Figure 3: Upper panel: Scatter plots of SMAP estimated soil moisture versus *in situ* soil moisture estimates by CRNS at the different field sites at Voulund (agriculture), Harrild (heathland) and Gludsted (forest) and as a weighted areal mean. Lower panel: Scatter plots of SMAP estimated soil moisture versus *in situ* soil moisture estimates by capacitance probes for different lad use types and he mean. Black dotted line shows the linear regression and grey dotted line the 1:1 line. $R^2$ and RMSE between soil moisture derived by SMAP and CRNS or capacitance probes are indicated in each subplot.**

### 3.1.2 Sentinel-1

### 3.1.2.1 LUC definition based on k-means Cluster analysis of Sentinel-1

The results of the k-means cluster analysis show that the different clusters have their distinct behaviours in mean and standard deviation of VV and VH, in space (Fig. 4) and time (Fig. 5). These groups can be related to the dominant LUC types, agriculture, heathland and forest. Cluster 1 (light grey) shows a medium mean-VV, high std.—VV, low-medium mean-VH and high std.-VH (Fig. 4). This cluster shows the most dynamics in mean VV and mean VH, i.e. lowest value in summer and has a much larger spatial variability than the other two clusters shown by the high std. (Fig. 5). This cluster is characterized by a high seasonal variability in biomass and soil moisture. The cluster corresponds to agricultural land use (compared to the ortophoto and LUC map) where the biomass changes with growing and harvest season and soil moisture is varying with precipitation and irrigation. Cluster 2 (grey) shows low mean-VV, low std.-VV, low mean-VH and low std.-VH (Fig. 4). Here the vegetation is low with little seasonal variation. This cluster corresponds to heathland which is also visible in comparison to the ortophoto and LUC map. In heathland and agriculture the mean VH shows similar temporal dynamics as mean VV with a clear decrease in summer. Cluster 3 (black) is characterized by high mean-VV, low std.-VV, high mean-VH and low std.-VH (Fig.4) and low seasonal variation (Fig. 5). This corresponds to forest with high amount of biomass which mask changes in soil moisture and with little seasonal variation due to the constant vegetation cover of pine trees. It seems unlikely that Sentinel-1 data carry information on soil moisture dynamics in the forest which is dominated by the biomass signal. For these areas the downscaled soil moisture estimates would mimic the one from SMAP, maybe except for the extraordinary dry summer of 2018 (Fig. 5). The different aggregation levels of the cluster analysis show that a resolution of 1000 m is quite coarse and consequently a lot of information is lost. Table 1 shows the fraction of LUC types for the three clusters at three different resolutions and for the LUC map. Compared to the LUC map, the fraction of heathland and forest are overestimated while the agriculture is underestimated. In the area the agricultural fields are surrounded by windbreaks which are classified by the k-means clustering as heathland or forest, see Fig. 4.

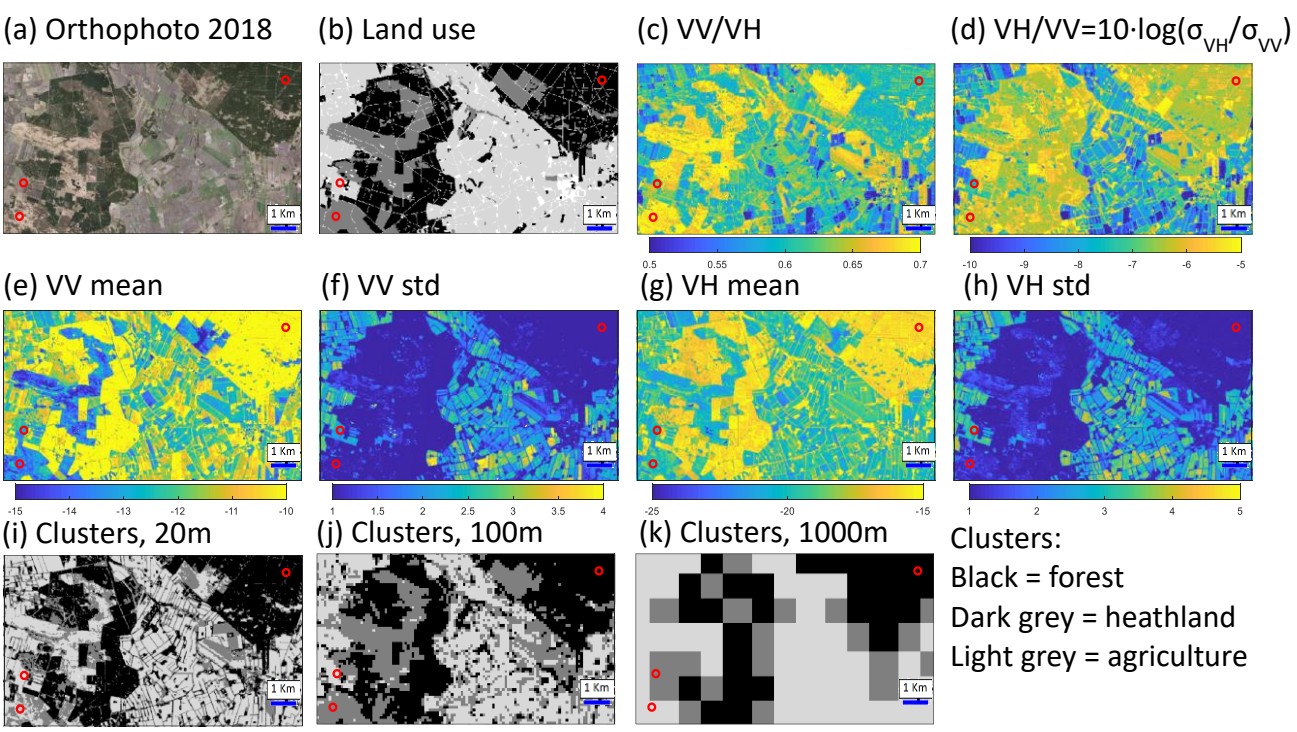

**Figure 4: Top: (a)** Ortophoto (GeoDanmark-data, 2018) of the study area (as in Fig. 1, c). **(b)** Land use map (Aarhus University, 2016) in a resolution of 10 m, light grey= agriculture, grey= heathland, black=forest, white cities and roads. **(c)** Cross-ratio in [dB]. **(d)** Cross-ratio in original backscatter. **(e)-(h)** Mean and standard deviation of Sentinel-1 co-and cross-correlation images at a 20 m scale. **(i)-(k)** Clustering results at different scales (20 m, 100 m and 1000 m). Locations of CRNS stations indicated in red circles on all subfigures.

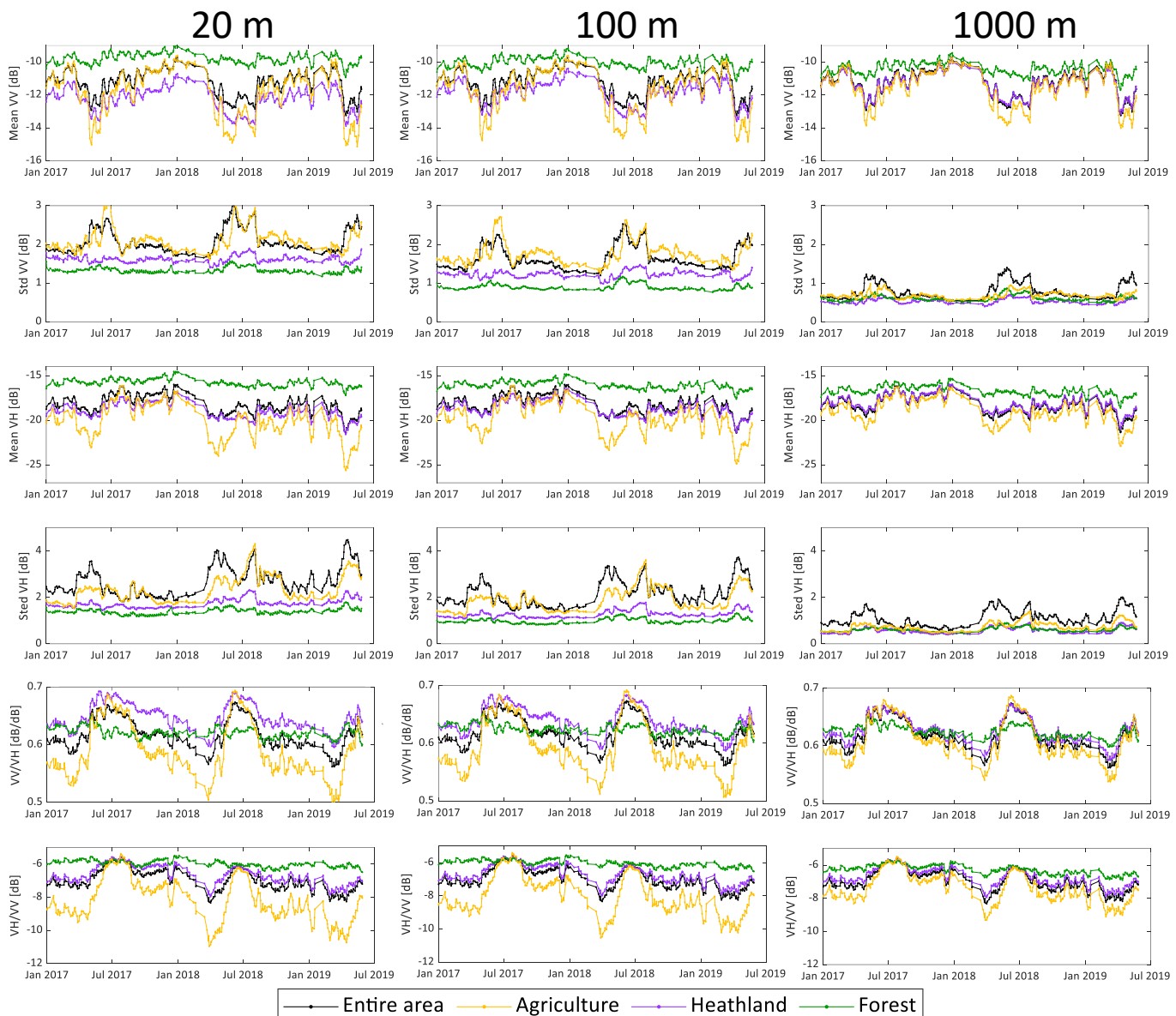

**Figure 5: Temporal dynamics of mean and standard deviation of VV, VH and the two cross ratios VV/VH and VH/VV for the different clusters at spatial resolutions of 20 m, 100 m and 1000 m.**

**Table 1: Fraction of LUC types in the LUC map and in the clusters at spatial resolutions of 20 m, 100 m and 1000 m.**

| Type | LUC map | 20 m | 100 m | 1000 m |
|---|---|---|---|---|
| Cities and lakes, etc. | 0.140 | -- | -- | -- |
| Agriculture | 0.557 | 0.399 | 0.349 | 0.430 |
| Heathland | 0.087 | 0.295 | 0.337 | 0.366 |
| Forest | 0.214 | 0.302 | 0.314 | 0.204 |

Comparing the cluster results at different resolutions with the known actual LUC types at the 30 capacitance probe locations show reasonable agreement for resolutions between 20 m-400 m with misclassifications of <25%. At these high resolutions, wrongly classified stations are typically located in the periphery of an agricultural field and are therefore classified as heathland (read windbreaks) instead of agriculture. There are also a few stations in the heathland wrongly classified as agriculture because large areas of the heathland are covered with grass. At coarser resolutions (1000 m) the misclassification is >42%. Both from the spatial pattern and classification statistics a spatial resolution higher than 1000 m is desirable.

To better understand the influence of biomass on the backscatter, the temporal dynamics of the cross-ratios are explored. VH/VV remains relatively constant without seasonal changes for the forest cluster (Fig. 5). For the heathland and particularly for the agriculture clusters VH/VV are more dynamic in time with higher values in summer and lower values in the winter. In winter the biomass is low in heathland and agriculture but relatively constant in the evergreen pine forests. VH/VV increases in spring when the growing period starts, reaches the maximum in summer (here VH/VV of agriculture and heathland reach a similar level as the forest), and decreases again after harvest in the autumn. The similar value observed across LUC types in summer could indicate that the signal might be more influenced by biomass than in the other seasons.

The variability across the area in VV and VH increases as the mean VV and VH values decreases as can be seen from plotting std. against mean values (Fig. 6). At resolutions of 20 m and 100 m the clusters are clearly identifiable by this relation, while at 1000 this is not the case, suggesting that 1000 m resolution is too coarse. At this scale, due to the heterogeneity of the area, LUC types start to overlap and therefore cannot be as clearly separated by the cluster analysis any more.

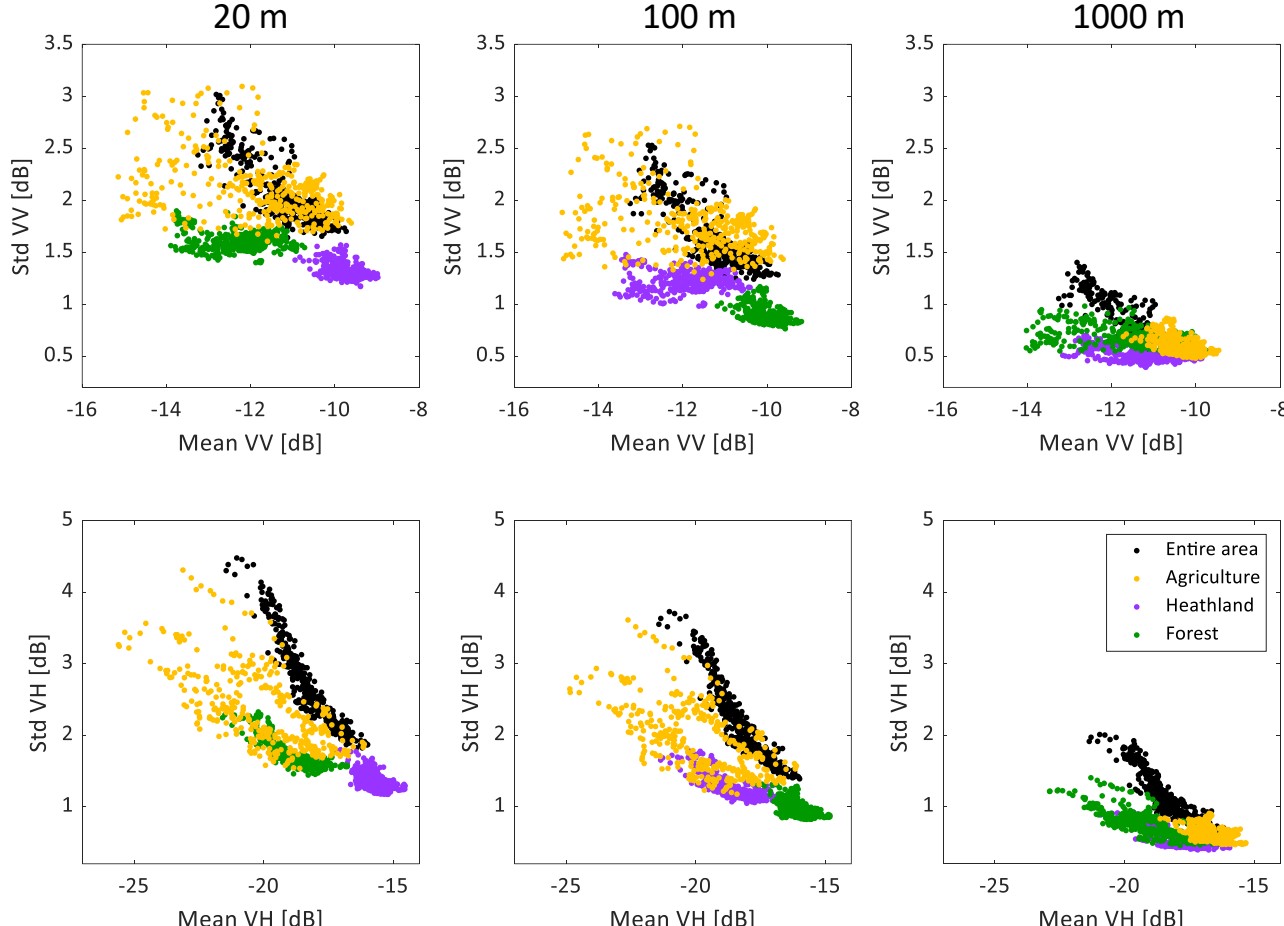

**Figure 6: Results of cluster analysis: relation between std. and mean for both VV and VH at three different spatial resolutions. Black is the relation for the entire area (cluster independent) and coloured are the respective relations for each cluster.**

The zoom into the agricultural field at Voulund (Fig. 7) shows that Sentinel-1 backscatter in VV, VH and their cross-ratio VH/VV aligns well with the dominant land cover types and that surrounding features, like windbreaks and other agricultural fields can be distinguished clearly as they have a different vegetation, management and irrigation schemes. These local heterogeneities are best kept at high resolutions and diminish when lowering the resolution as shown by the clusters at different spatial resolutions.

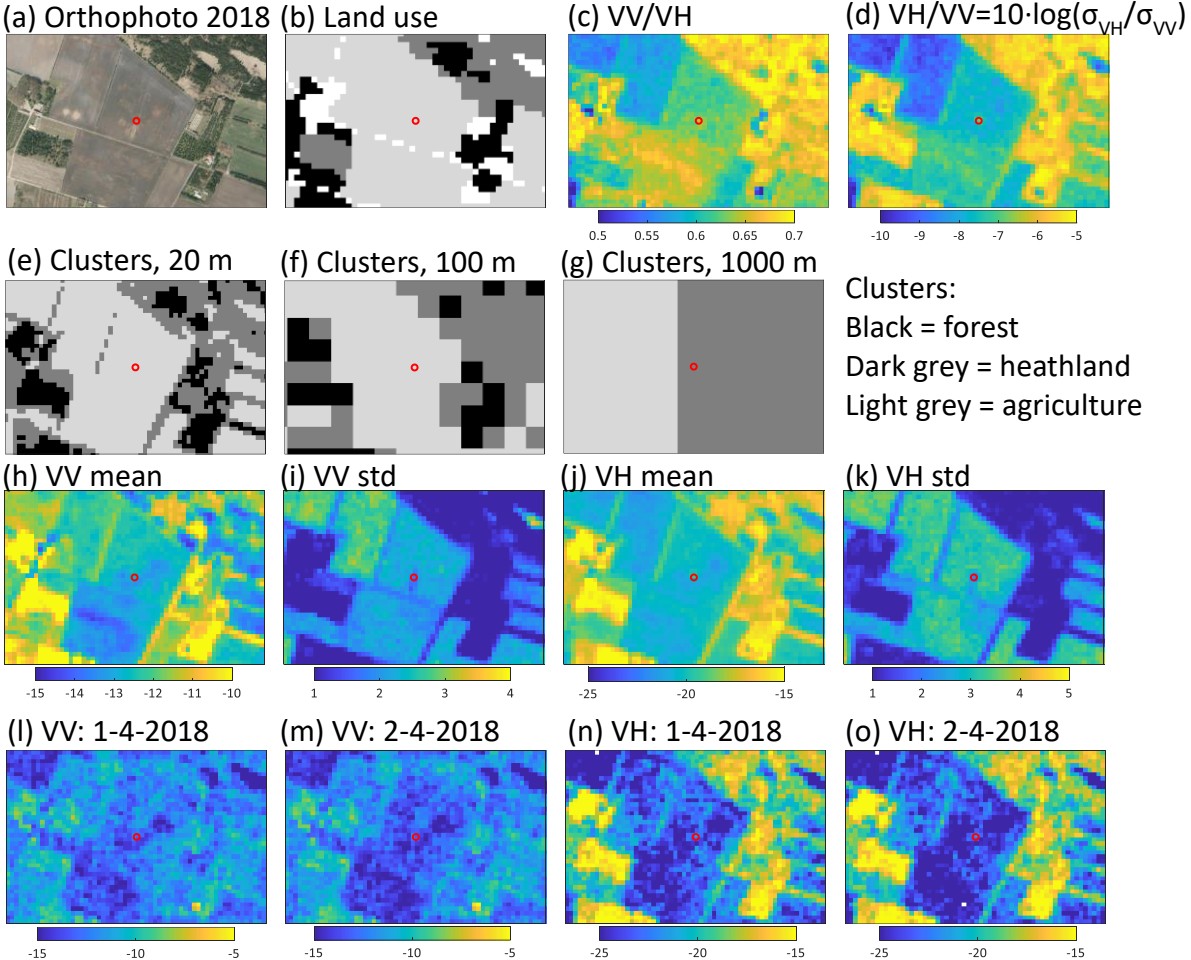

**Figure 7: Zoom on the agricultural field site Voulund at a resolution of 20 m. (a) Ortophoto (GeoDanmark-data, 2018), (b) land use map (Aarhus University, 2016), (e)-(g) Clusters maps at 20 m, 100 m and 100 m spatial resolution. (l)-(o) Spatial pattern of Sentinel-1 backscatter in VV, VH , at single dates during the year 2018 as well as the respective mean and standard deviation of the entire period (h)-(k) and (c) VV/VH and (d) VH/VV. Red circle indicates the location of the CRNS station at Voulund.**

### 3.2 Downscaling resolution

To identify the optimal downscaling resolution VV and VH have been aggregated to different scales at the three CRNS sites, 20m, 50m, 100m, 200m, 400 m and 1000 m (for brevity, only results for 20 m, 100 m and 1000 m are shown on Fig. 8). The temporal dynamics at the different aggregation scales were evaluated based on how noisy they appear, how much they still allow distinguishing seasonal signals and on how much they are influenced by mixed LUC types that result in smoothing out the signal. Time series of VV and VH and backscatter at different resolutions at the three different study sites ((Fig. 8) show that the signal loses a considerable amount of temporal dynamics and hence a lot of information at a level of 1000 m resolution. At a resolution of 20 m the signal on the other hand is rather noisy and a meaningful resolution that represents the seasonal variation without much noise is to be expected in the scale of hundreds of meters.

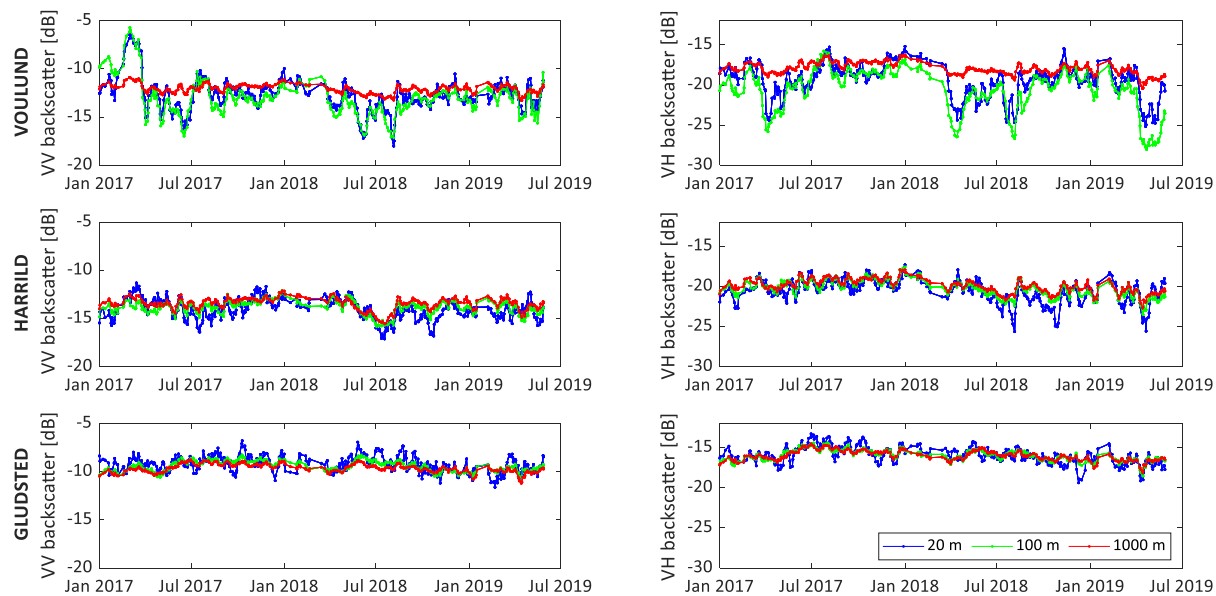

**Figure 8: Time series of Sentinel-1 backscatter, VV, VH at different aggregation levels (20 m, 100 m and 1000 m) at the three CRNS sites at Voulund (agriculture), Harrild (heathland) and Gludsted (forest).**

The soil moisture derived by CRNS shows a good linear correlation with Sentinel-1 VV and VH backscatter at a resolution of 100 m and 200 m at the agricultural and heathland sites (Table 2, Supplemental Material Fig. S2). At Harrild (heathland) the correlation is highest due to the minimal changes in vegetation cover of the heathland. There are other factors apart from soil moisture that control VV which becomes clear at the agricultural site where the vegetation cover changes over the seasons. At the forest site (Gludsted) where the VV backscatter is dominantly influenced by biomass there is almost no signal of soil moisture and hence the correlation between CRNS and VV is poor. Somewhat surprising is the correlation of VH and CRNS derived soil moisture at Voulund and Harrild, indicating that at resolutions <1000 m soil moisture impacts VH backscatter.

**Table 2: Statistics of linear regression between VV and VH backscatter at different spatial resolution at the specific HOBE locations and CRNS derived soil moisture. See the corresponding scatterplots in the supplemental material (Fig. S2).**

| spatial resolution | **VOULUND (agriculture)** | | **HARRILD (heathland)** | | **GLUDSTED (forest)** | |
|---|---|---|---|---|---|---|
| | $R^2$ | | $R^2$ | | $R^2$ | |
| CRNS versus | VV | VH | VV | VH | VV | VH |
| 20 m | 0.267 | 0.430 | 0.353 | 0.309 | 0.018 | 0.016 |
| 50 m | 0.254 | 0.336 | 0.376 | 0.346 | 0.004 | 0.006 |
| 100 m | 0.292 | 0.290 | 0.477 | 0.469 | 0.005 | 4.1e-06 |
| 200 m | 0.281 | 0.313 | 0.568 | 0.619 | 5.1e-04 | 0.005 |
| 400 m | 0.286 | 0.273 | 0.465 | 0.392 | 1.4e-04 | 2.5e-04 |
| 1000 m | 0.613 | 0.369 | 0.571 | 0.451 | 9.5e-04 | 8.9e-04 |

### 3.3 Downscaling soil moisture using the SMAP Sentinel-1 approach

As we just showed, the Sentinel-1 backscatter shows a good correlation to ground-based soil moisture observations in the heathland and agriculture at resolutions <1000 m. Therefore a downscaling approach using the SMAP Sentinel-1 approach to derive soil moisture pattern at the 100 m scale is pursued in areas used for agriculture or

covered by heathland, corresponding to approximately 2/3 of the study area. The SMAP Sentinel-1 downscaling approach involves the estimation of two parameters β and Γ.

The assumption that β is invariant in time and space is not totally valid as C-SAR is influenced by vegetation cover which changes both over time (growing/harvesting season) and space due to LUC (e.g., grass and forest). However, a time invariant β is applied because the vegetation and surface roughness barely change in the heathland and evergreen pine forest. There is a change in vegetation and roughness in the agriculture due to land management. However, the crop yield is relatively constant and changes in biomass relatively small Andreasen et al. (2020).

Moreover, time series of VV and VH (e.g. Figure 5) show the opposite trend as would be expected if the backscatter signal would be dominated by vegetation and surface roughness. A higher amount of vegetation would enhance the volumetric backscatter, both for co- and cross polarization (e.g. Rosenqvist, 2018) but what can be observed is a reduction of the backscatter signal in the growing and peak vegetation periods (spring and summer). On the other hand, in these periods soil moisture is low due to relatively high temperature and evapotranspiration. This

trend is mostly observed in the agriculture, while less visible in the heathland and almost not significant in the forest. An evaluation whether a seasonally varying β-estimation would be essential to consider (details can be found in the Supplemental Material S3) showed that a time invariant β seems to be a suitable simplification for the present study site in a typical Danish environment.

To take the spatial variability of β but also Γ into account, we additionally explored varying these downscaling parameters in space according to the LUC clusters (see Table 3; Supplemental Material Fig. S4 and Table S4). The cluster dependent β was estimated as the linear regression of the time series $\frac{\theta_{SMAP}}{VV_{cluster\ mean}}$. Hereby $VV_{Cluster\ mean}$ is the time series of spatial mean of the co-polarized backscatter signal (VV) of all Sentinel pixels in each cluster within the corresponding SMAP pixel. The distribution of the clusters and the mean and std. of the backscatter signal are shown in Figure 4. Finally, the impact of including a temporal Γ was explored. In the current

study a temporal window of 40 data points was used.

**Table 3: Statistics of β (θ vs. VV) estimation at spatial resolutions of 20 m, 100 m and 1000 m, shown for β invariant in space (entire area) or variant in space according to the three clusters.**

| β-estimation | 20 m | | 100 m | | 1000 m | |
|---|---|---|---|---|---|---|
| | β [m³/m³/dB] | $R^2$ | β [m³/m³/dB] | $R^2$ | β [m³/m³/dB] | $R^2$ |
| Entire area | 0.074 | 0.693 | 0.074 | 0.694 | 0.074 | 0.693 |
| Agriculture | 0.041 | 0.672 | 0.044 | 0.683 | 0.056 | 0.702 |
| Heathland | 0.083 | 0.721 | 0.082 | 0.722 | 0.081 | 0.690 |
| Forest | 0.128 | 0.346 | 0.128 | 0.364 | 0.125 | 0.443 |

    Downscaled soil moisture maps were produced at spatial resolutions of 20 m, 100 m, and 1000 m. For conciseness we present the results of the 100 m and 1000 m resolution (Fig. 9 and Supplement Material Fig. S6 and S7). Eight types of β and Γ combinations (Table 4) were used and evaluated against the ground-observed soil moisture. Hereby, β was either estimated invariant in space and time as $\theta_{SMAP}/VV$ or spatial varying according to the three

LUC as $\theta_{SMAP}/VV_{Cluster}$ (compare Table 4 and Supplemental Material Fig. S4). Γ was estimated either as invariant in space and time as $\delta VV_{mean}/\delta VH_{mean}$, invariant in time but varying according to the three LUC as $\delta VV_{Cluster\_mean}/\delta VH_{Cluster\_mean}$, time-varying (space invariant) as $\delta VV_{mean}/\delta VH_{mean}$ (applying a moving window of 40) or time and space varying estimated as $\delta VV_{Cluster\_mean}/\delta VH_{Cluster\_mean}$ (applying a moving window of 40) (Table 4).The evaluation of the downscaling results is based (1) on the statistical values of the non-linear least square

regression of time series of downscaled and CRNS derived soil moisture (slope, $R^2$, and RMSE) at the three field sites and (2) on the minimized order difference of downscaled soil moisture and the capacitance probe network located in the heathland (three stations) and in the agriculture (thirteen stations). Moreover, a similar or reduced RMSE as between original SMAP and CRNS (RMSE ≈ 0.056 [-], Fig. 3) is evaluated positively. Since the original coarse resolution SMAP fits very well and contains the average soil moisture information, a high reduction in

RMSE when comparing temporal dynamics is not expected. Therefore, in the present study, the additional value

of the downscaling task lies more in the improved spatial resolution than in the better statistical fit of already well matching time series.

**Table 4: Combinations of the downscaling parameters β and Γ applying the classical algorithm with one parameter for the entire area (indicated in bold) and different types applying cluster dependent parameters (indicated in *italic*).**

| Combinations of β and Γ (types) | β [m³/m³/dB] | Γ [dB/dB] |
|---|---|---|
| | Slope of θ$_{SMAP}$/VV | δVV/δVH |
| Type 1 | **1 constant β** | **1 constant Γ** |
| Type 2 | **1 constant β** | *3 constant Γ* |
| Type 3 – classic | **1 constant β** | **1 time-varying Γ** |
| Type 4 | **1 constant β** | *3 time-varying Γ* |
| Type 5 | *3 constant β* | **1 constant Γ** |
| Type 6 | *3 constant β* | *3 constant Γ* |
| Type 7 | *3 constant β* | **1 time-varying Γ** |
| Type 8 | *3 constant β* | *3 time-varying Γ* |

At the 1000 m spatial resolution the classical approach (type 3) produces acceptable results (Table 5). The downscaled soil moisture fits very well with the CRNS derived soil moisture at the agricultural site (Fig. 10, a shows results for type 8 but type 3 time series are alike). At the heathland the downscaled soil moisture time series mostly follows the original SMAP signal. Here a time-lag is observed when it dries out in the summer of 2018.
The RMSE of the least square regression between downscaled and CRNS derived soil moisture improves by around 50% to 0.034 [-] at the agricultural field compared to the original CRNS versus SMAP RMSE of 0.072 [-]. At the heathland the RMSE gets worse from 0.065 [-] (CRNS versus SMAP) to 0.121 [-] for the downscaled soil moisture while at the forest site the RMSE stays in the similar range to the original one. At the 1000 m scale these statistics do not significantly improve by using a cluster dependent β or cluster dependent, time varying Γ.
The mSDR of the capacitance probes at the 1000 m scale ranges between 5.6 and 6.1. The spatial pattern of mean downscaled soil moisture (Fig. 9) show low values in the heathland, medium values in the forest and medium to high values in the agriculture. The standard deviation of the downscaled soil moisture is low in heathland and forest and higher in the agriculture. The downscaling type 4 enhances the differences in std θ between heathland and forest (low) compared to agriculture (high) while type 8 diminishes these (Fig. 9).

**Table 5: Statistics of downscaling results: downscaled soil moisture vs. CRNS derived soil moisture at the different filed sites; difference in rank of capacitance probes.**

| | CRNS at VOULUND (agriculture) | | | CRNS at VOULUND (shorter period) | | | CRNS at HARRILD (heathland) | | | CRNS at GLUDSTED (forest) | | | Capacitance probes |
|---|---|---|---|---|---|---|---|---|---|---|---|---|---|
| | slope | R² | RMSE [-] | slope | R² | RMSE [-] | slope | R² | RMSE [-] | slope | R² | RMSE [-] | mSDR |
| 100 m | | | | | | | | | | | | | |
| Type 1: | 1.170 | 0.189 | 0.124 | 0.902 | 0.305 | 0.073 | 0.598 | 0.345 | 0.138 | 0.548 | 0.289 | 0.061 | 5.9 |
| Type 2: | 1.501 | 0.307 | 0.123 | 1.190 | 0.469 | 0.107 | 0.614 | 0.352 | 0.153 | 0.331 | 0.185 | 0.111 | 6.0 |
| Type 3: | 1.014 | 0.139 | 0.130 | 0.781 | 0.292 | 0.067 | 0.508 | 0.346 | 0.136 | 0.662 | 0.301 | 0.068 | 6.3 |
| Type 4: | 1.472 | 0.283 | 0.126 | 1.181 | 0.479 | 0.106 | 0.542 | 0.355 | 0.151 | 0.322 | 0.177 | 0.112 | 5.6 |
| Type 5: | 0.835 | 0.236 | 0.098 | 0.709 | 0.350 | 0.065 | 0.596 | 0.314 | 0.104 | 1.148 | 0.398 | 0.173 | 4.4 |
| Type 6: | 1.030 | 0.365 | 0.071 | 0.879 | 0.518 | 0.045 | 0.613 | 0.335 | 0.118 | 0.772 | 0.394 | 0.069 | 3.9 |
| Type 7: | 0.743 | 0.185 | 0.103 | 0.638 | 0.339 | 0.063 | 0.497 | 0.307 | 0.101 | 1.346 | 0.369 | 0.194 | 5.8 |
| Type 8: | 1.013 | 0.342 | 0.074 | 0.874 | 0.532 | 0.043 | 0.534 | 0.334 | 0.115 | 0.757 | 0.383 | 0.070 | 4.5 |
| 1000 m | | | | | | | | | | | | | |
| Type 1: | 1.013 | 0.724 | 0.033 | 1.003 | 0.727 | 0.035 | 0.659 | 0.412 | 0.123 | 0.619 | 0.426 | 0.057 | 5.6 |
| Type 2: | 0.975 | 0.701 | 0.032 | 0.971 | 0.700 | 0.033 | 0.744 | 0.412 | 0.158 | 0.411 | 0.331 | 0.070 | 5.6 |
| Type 3: | 1.030 | 0.731 | 0.034 | 1.014 | 0.731 | 0.035 | 0.579 | 0.412 | 0.121 | 0.731 | 0.416 | 0.065 | 5.8 |
| Type 4: | 0.977 | 0.708 | 0.032 | 0.969 | 0.712 | 0.033 | 0.687 | 0.401 | 0.161 | 0.415 | 0.329 | 0.070 | 5.8 |
| Type 5: | 1.084 | 0.740 | 0.036 | 1.063 | 0.740 | 0.037 | 0.541 | 0.383 | 0.088 | 1.173 | 0.541 | 0.142 | 6.1 |
| Type 6: | 1.041 | 0.722 | 0.033 | 1.028 | 0.726 | 0.035 | 0.606 | 0.398 | 0.112 | 0.821 | 0.608 | 0.043 | 6.0 |

| | | | | | | | | | | | | |
|---|---|---|---|---|---|---|---|---|---|---|---|---|
| Type 7: | 1.104 | 0.745 | 0.037 | 1.075 | 0.740 | 0.038 | 0.481 | 0.379 | 0.088 | 1.361 | 0.476 | 0.162 | 5.9 |
| Type 8: | 1.043 | 0.729 | 0.033 | 1.025 | 0.729 | 0.034 | 0.562 | 0.393 | 0.114 | 0.829 | 0.600 | 0.043 | 6.0 |
| | | | | | | | | | | | | | |
| CRNS vs. original SMAP | 1.246 | 0.710 | 0.072 | 1.24 | 0.714 | 0.074 | 0.708 | 0.408 | 0.065 | 0.890 | 0.619 | 0.061 | |

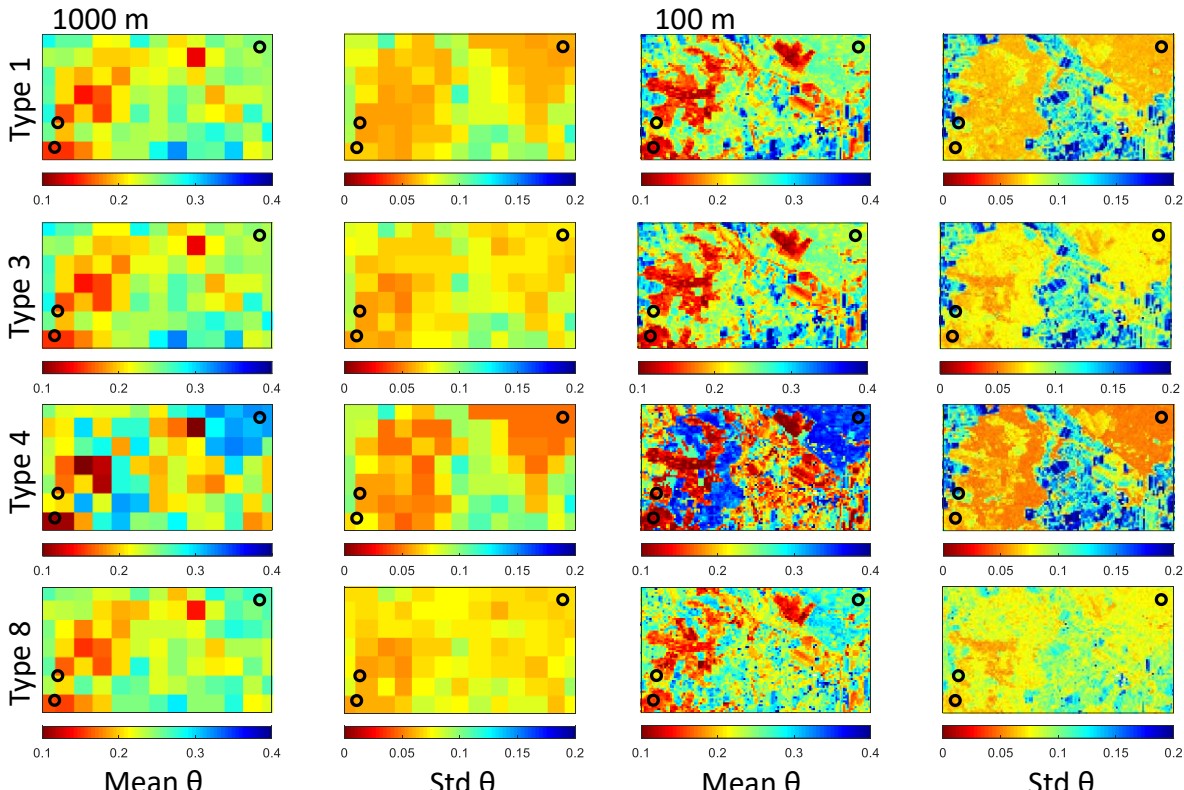

**Figure 9: Downscaling results at the 1000 m and 100 m spatial resolution: spatial pattern of the mean and standard deviation of downscaled soil moisture shown for types 1, 3, 4 and 8.**

At the 100 m spatial resolution the statistics of the downscaled and CRNS derived soil moisture (Table 5) are almost equally good for the different types and at each of the CRNS locations. The statistical results at Voulund (Table 5) are shown for the whole period and a shorter period (12th May 2017 to 31st January 2018 and 1st April 2018 to 29th March 2019) as there are some obvious mismatches in the winter/spring seasons that overrule the otherwise good match between CRNS and downscaled soil moisture. The $R^2$ are around 0.3 at Harrild (heathland),
and 0.2 to 0.4 at Gludsted (forest) and the RMSE are around 0.10 [-] to 0.15 [-] at Harrild and more varying (between 0.06 [-] to 0.191 [-]) at Gludsted. At the agricultural site $R^2$ is between 0.13 and 0.37 for the whole period and between 0.29 and 0.53 for the shorter period while the RMSE is 0.07 [-] to 0.13 [-] and 0.04 [-] to 0.11[-] for the respective periods. The mSDR of capacitance probes ranges between 6.3 and 3.9 for all types. The downscaling soil moisture results at the 100 m scale improve with a LUC dependent β and Γ. However, the spatial distribution
of downscaled soil moisture differs substantially between the different types (Fig. 9).

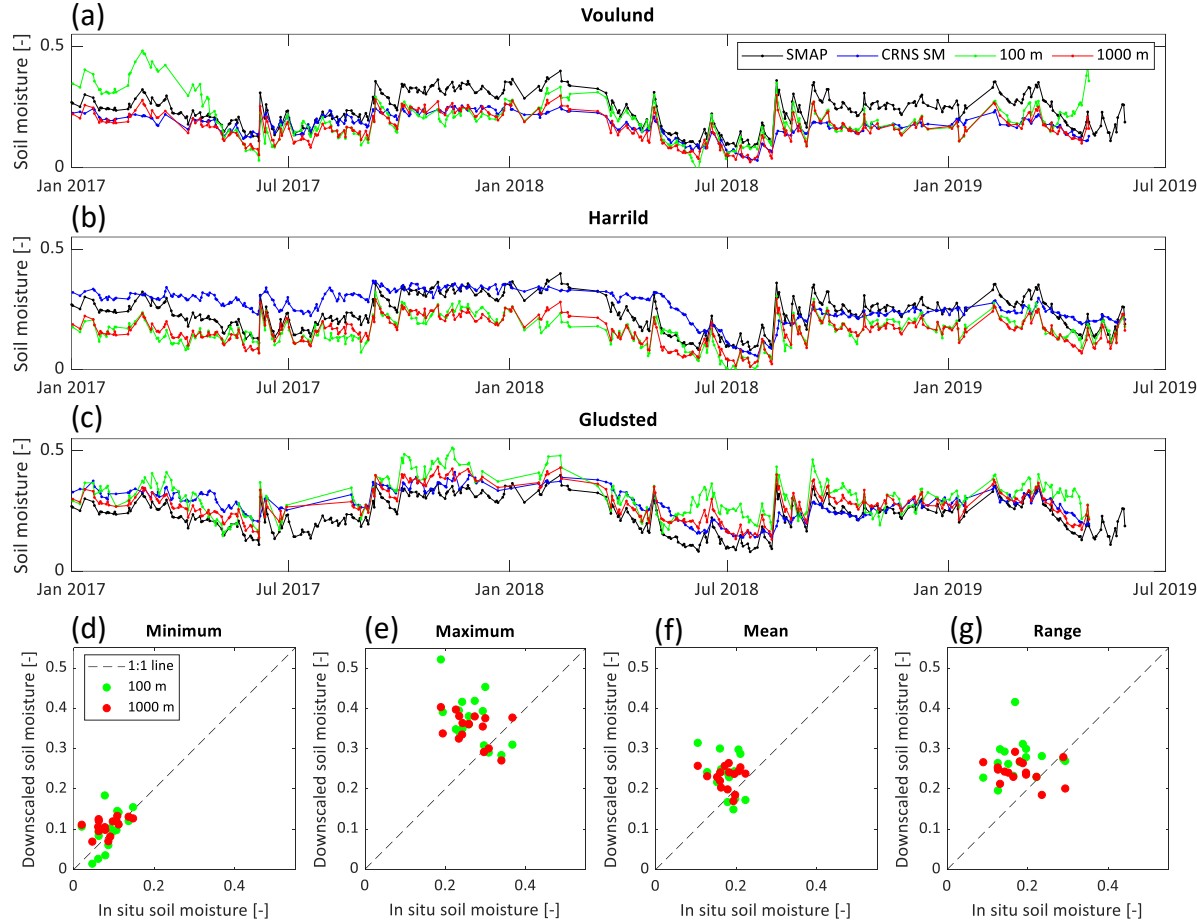

**Figure 10: (a) – (c): Temporal dynamics of SMAP soil moisture (averaged over the 21 pixels),** *in situ* **soil moisture (CRNS) and downscaled soil moisture (type 8) at 100 m and 1000 m spatial resolution at the three different CRNS sites at Voulund (agriculture), Harrild (heathland) and Gludsted (forest). (d) – (g): Scatter plot of minimum, maximum, mean and range of downscaled soil moisture at 100 m and 1000 m spatial resolution versus** *in situ* **soil moisture by capacitance probes.**

## 4. Discussion

From the design of the present study a thorough analysis of SMAP and Sentinel-1 data compared to local estimates of soil moisture by CRNS and capacitance probes was conducted before we applied the SMAP Sentinel-1 downscaling algorithm to produce distributed soil moisture pattern at high spatial resolutions of 20 m and 100 m. The novelty of our approach stems from (1) the downscaling resolution with goes far below 1 km which has been the lower target resolution of many recent studies (2) vegetation dependent parameters in downscaling algorithm and (3) the combination/validation of downscaled soil moisture with a variety of ground-based soil moisture observation, i.e. capacitance probes as well as stationary CRNS. The latter method allows estimating soil moisture at a scale of a few hundred meters which compares with the target resolution of the downscaled soil moisture.

### 4.1 Data analysis

SMAP derived soil moisture and Sentinel-1 backscatter at different spatial resolutions were compared to *in situ* observations of soil moisture by CRNS and capacitance probes distributed over different LUC classes. The application of CRNS soil moisture estimates complementing the more conventional capacitance probes measurement constitutes a significant improvement for validation of remotely sensed and downscaled soil moisture products. The major advantage is the similar scale of horizontal sensitivity of the downscaled product and the CRNS which is in the order of few hundreds meter. However, the differentiation between soil moisture

signal and noise constitutes still a major challenge. To minimize the impact of noise, the Sentinel backscatter was smoothed spatially (a minimum aggregation to 20 m) and temporally, by applying a moving average of five images. Moreover, meaningful trends in the time series are visible that follow the season and the expected soil moisture. By validating against different independent ground measurements (CRNS and capacitance probes) the reliability of the approach and the obtained results is enhanced. However, each of the independent measurements has advantages and disadvantages. In our study area the capacitance probe time series had many gaps and therefore we complemented it by using CRNS which in turn are subject to uncertainties related to soil water content and sensing footprint (as discussed in paragraph 2.3.4).

Soil moisture estimates derived by SMAP (0.08 – 0.4) show a higher dynamic range than the one derived by CRNS (0.08 - 0.3) which might be caused by the more surficial penetration depth of the SMAP (Mohanty et al., 2017). Similar to Peng et al. (2017), we therefore believe that the dynamics are more reliable than absolute values. The temporal dynamics between the two correlate well for the entire area ($R^2$= 0.7, RMSE = 0.056 [-]) and at the agricultural site (Voulund, $R^2$= 0.71, RMSE = 0.072 [-]) while a little bit less for the heathland (Harrild, $R^2$= 0.408, RMSE = 0.065 [-]) and forest (Gludsted, $R^2$= 0.619, RMSE = 0.061 [-]) sites. Similarly, the SMAP estimated soil moisture and the mean of the capacitance probes network are in accordance ($R^2$= 0.557, RMSE = 0.048[-]) which was expected since the capacitance network is used for validation of the SMOS mission (Bircher et al., 2012). However, the single capacitance probe locations (some have a poor data coverage) diverge significantly from the mean and the SMAP due to local heterogeneities in soil properties that are not disintegrated on the coarse spatial resolution of SMAP. The correlation between SMAP and CRNS is better than between SMAP and capacitance probes (cf. Fig. 3) which supports the advantage of observations at more comparable spatial resolutions. The downscaling task adds to further improve the comparability of the remotely sensed and *in situ* observed soil moisture.

Spatial changes in Sentinel-1 backscatter are influenced by the dominant LUC, and have been used for a k-means clustering. Temporal changes for each cluster are significantly different, regarding both absolute mean values, dynamic behavior and standard deviation. Changes in biomass appears to be best described using the cross-ratio VH/VV [dB] which has also been suggested by Harfenmeister et al. (2019). The analysis of an appropriate downscaling scale showed that lower resolutions cause increased misclassification of LUC and that the temporal changes in Sentinel-1 at specific sites are reduced markedly. A downscaling resolution of 100 m is therefore desirable. A good correlation between CRNS and Sentinel-1 backscatter at 100-200 m resolution is observed at the low biomass locations (Harrild and Voulund) (as expected due to the footprint of the CRNS and the highly heterogeneous agricultural landscape with filed sizes in the scale of hectometers (<1000m$^2$)). From the combined time series analysis it becomes clear that the Sentinel-1 signal in the forest is very much influenced by vegetation structure and no or little correlation can be drawn to soil moisture. One explanation could be the wavelength of the C-SAR backscatter of 5.6 cm which is best in detecting objects of similar sizes (Rosenqvist, 2018). Therefore, the C-band is sensitive to soil moisture within sparse and low biomass areas as in the heathland and in the agriculture while the signal is dominated by volumetric scattering of the pine trees in the forest. In the heathland, the vegetation is low and relatively constant over the seasons and dynamics in backscatter relate therefore to changes in soil moisture. In the agriculture the amount of biomass (8.42t/ha) and its seasonal change are rather small (Andreasen et al., 2020) and hence the CRNS signal is mostly representative for soil moisture in this area. This is supported by the time series that show lower backscatter values in spring/summer during the growing and peak season. If the backscatter would be highly influenced by vegetation changes rather than soil moisture the opposite would be expected, a positive correlation between backscatter and vegetation (higher backscatter signals coinciding with higher vegetation). One exception might be the very high (less negative) VV backscatter at Voulund (e.g. Fig. 8) at the beginning of the study period which might be  rather an artefact as a result from agricultural soil management, e.g. tilling than a signal influenced by soil moisture.

### 4.2 Soil moisture dependency on Sentinel-1 backscatter

The similar temporal evolution of VV and VH at the heathland and agriculture clusters supports the assumptions that VH is not only impacted by biomass but also to a significant extent by soil moisture. This has been reported previously (El Hajj et al., 2019; Harfenmeister et al., 2019). Comparing VV and VH backscatter with the CRNS derived soil moisture at the different field sites (Table 2; Supplemental Material Fig. S2) shows that at high spatial resolutions (20 m - 400 m) VH shows higher or at least as much correlation as VV with soil moisture at Voulund (agriculture) and Harrild (heathland). At the Gludsted forest site, the correlation for both VV and VH to CRNS soil moisture is very poor at all spatial resolutions (Table 2) possibly due to the above mentioned influence of volume scattering of the leafage. Hence, the principal assumption that VH is primarily influenced by biomass while VV is influenced by biomass and soil moisture only holds for coarse spatial resolution of 1000 m and is possibly restricted to areas without dense forest.

### 4.3 Downscaling

The modified SMAP Sentinel-1 downscaling algorithm that has previously been successfully tested (He et al., 2018) was applied. At a first glance the results looked reasonable since the downscaling parameters β, Γ, as well as SM values were estimated within logical ranges and in the same order as in He et al. (2018) (β ranging from 0.03-0.093 [m³/m³/dB] and Γ ranging from 0.5 – 0.9). The performance statistics of $R^2$ and RMSE were acceptable and in ranges that were previously published (e.g. He et al., 2018). At Voulund (agriculture), the downscaling algorithm produces similar results at 100-m- and 1000-m-scale. The downscaled soil moisture time series appear trustworthy as the downscaled soil moisture is different from the original SMAP soil moisture and is in agreement to the measured CRNS soil moisture (Fig. 10, a). However, for the 100-m resolution there are still some discrepancies in soil moisture during late winter-early spring that could be related to field management practices. At Harrild (heathland), the downscaled soil moisture is very similar for the 100-m- and the 1000-m-scale (Fig. 10, b), indicating that the sub-kilometer downscaling is as trustworthy as the kilometer-scale. The downscaling algorithm has decreased the absolute values and the dynamic changes are dampened compared to the original SMAP soil moisture. However these changes are not in agreement with the CRNS soil moisture values, which in general have higher soil moisture throughout the year. Furthermore, the timing of the drying out summer 2018 is not fully captured. Moreover, from visual assessment, at the 100 m spatial resolution the downscaled spatial soil moisture pattern show a strong dependency on LUC types, particularly when applying the classical (type 3) algorithm (Fig. 9). These patterns look also very similar to the Sentinel-1 backscatter patterns (e.g. Fig. 4). It is questionable if the soil moisture depends to such a strong degree on the LUC, given that the area is affected by the same climatic effects and soil properties change independent from LUC. Therefore, it is also questionable if the downscaled soil moisture is correct. The cluster dependent downscaling scheme (e.g., type 8) dampens the strong LUC pattern and also minimizes the mismatch of the order of capacitance probes. Albeit, similar statistics (Table 5) can result in reversed soil moisture patterns, e.g. forest is very dry (e.g. type 5) or very wet (e.g. type 4) compared to the heathland and agriculture (Supplemental Material Fig. S7). Our study shows that the statement of González-Zamora et al. (2015) about the comparison of satellite derived soil moisture and *in situ* observations is also valid for the downscaled product. They concluded that temporal dynamics in soil moisture can be better reproduced than spatial patterns.

Our downscaling results underpin that the original SMAP soil moisture has a very good match in temporal dynamics with the ground observations; therefore, the statistics are difficult to improve significantly by downscaling using Sentinel-1. We showed that the spatial patterns change significantly with different small modifications to the classical downscaling algorithm and it remains challenging to clearly identify the best approach. Nevertheless we demonstrate that soil moisture estimates at the 1000 m and 100 m spatial resolution are improved with cluster dependent β. Additionally, on the 100 m scales, a cluster depend Γ improves the downscaled soil moisture results, which is not the case at the 1000 m scale.

## 5. Conclusion

Our study shows a strong correlation between Sentinel-1 VV, VH backscatter and CRNS soil moisture at the agricultural and heathland sites in central Denmark ($R^2$ in the magnitude of 0.3 to 0.6 depending on the different scales) and also a good match between SMAP and CRNS, particularly in temporal dynamics ($R^2$ =0.7, RMSE = 0.056 [-]). However, applying the well-established SMAP Sentinel-1 downscaling algorithm remains a challenge for higher spatial resolutions (20 m - 400 m). One reason is the strong correlation between VH and soil moisture at this scale, because the SMAP Sentinel-1 algorithm assumes that VH is predominantly influenced by biomass and vegetation structure. This seems valid for the coarser spatial resolutions (>1000 m) to which the algorithm has been successfully applied many times before. To dampen the otherwise close resemblance of downscaled soil moisture to LUC pattern we introduced LUC dependent downscaling parameters (β, Γ) which improved the results only marginally. Nevertheless, the soil moisture pattern of the downscaled product remained ambiguous. Since it is possible to create a well matching relationship of VV, VH and local CRNS further modifications to the algorithm are needed to solve the current challenge in downscaling to sub-kilometre spatial resolutions. To be successful in this endeavour, such analysis would benefit from a larger data set of *in situ* measurements at the relevant scale to better validate the spatial patterns produced. This could be achieved by expanding the study area and including more CRNS stations (e.g. there exists a network of about 50 CRNS stations across Europe (Bogena et al., 2022)). The use of such a network for calibration of the downscaling parameters β and Γ might be successful and hence improve the algorithm substantially.

**Acknowledgements**

This study was partly funded by the Danida Fellowship Center project 17-M10-KU. We greatly benefited from the rich data sets collected within the HOBE project.

We thank the editor Alexander Gruber and two anonymous reviewers for their profound reading of the manuscript and their constructive comments that we believe considerably improved the quality of this article.

**Funding**

This study was partly funded by the Danida Fellowship Center project 17-M10-KU.

**Declaration of interests**

The authors declare that they have no known competing financial interests or personal relationships that could have appeared to influence the work reported in this paper.

**Code/Data availability**

The remote sensing data is freely available from NASA (SMAP; https://search.earthdata.nasa.gov) and ESA (Sentinel-1, we retrieved the Copernicus S1_GRD data set via google earth engine). The capacitance probes ground observations are part of the International Soil Moisture Network and can be retrieved from their website https://ismn.earth/en/networks/?id=HOBE. The CRNS data is part of the COSMOS-Europe network which has been recently presented in an ESSD data paper (https://doi.org/10.5194/essd-14-1125-2022).

**Author's responsibilities**

RM, WZ, KHJ, RF, SS, MCL designed the study. RM, WZ, SJK, MCL and MA prepared the data and participated in the preliminary data analysis. RM and MCL carried out the downscaling analysis and interpretation and designed the figures. RM took the lead in writing with input from MA and in close consultation with MCL. All authors discussed results and provided critical feedback to the manuscript drafts and approved the final version of the manuscript.

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
