# Peer review of "Title: Exploring the combined use of SMAP and Sentinel-1 data for downscaling soil moisture beyond the 1 km scale"

_Hydrology and Earth System Sciences, 2021_

## Referee Comment (RC1)

Exploring the combined use of SMAP and Sentinel-1 data for downscaling soil moisture beyond the 1 km scale

The authors test the possibility of downscaling SMAP coarse soil moisture to the sub-kilometer resolution using Sentinel-1 SAR data. This paper is interesting and the topic is suitable to HESS. However, I have several major comments the authors should seriously consider.

Major comments:

1. This paper directly disaggregates the SMAP coarse soil moisture at 9 km to high resolution using Sentinel-1 SAR backscattering coefficients. It should be noted that the method tested in this paper is based on the assumption of a near-linear relationship between radar backscatter $\sigma^0_{pp}$ and soil moisture $\theta$ at different scales. In order to estimate the parameter $\beta$, a time regression is performed under the assumption that the soil roughness and vegetation conditions do not change greatly over a specified temporal window. Meanwhile, the parameter $\beta$ is NOT invariant in time and space and it depends on vegetation cover and type as well as surface roughness. Therefore, a moving window of $\beta$ estimation should be adopted when applying this downscaling algorithm to a long time period and the length of time window should be carefully determined. In this study, about 377 images of synchronized SMAP and Sentinel-1 were obtained during the period of January 1, 2017 to May 31, 2019. However, this paper did not describe how to determine the parameter $\beta$. In Page 15 Line 445, a temporal window of 40 data points was used to derive seasonal $\Gamma$. To derive $\beta$?

2. Page 14 Line 425-426: The soil moisture derived by CRNS shows a good linear correlation with Sentinel-1 VV and VH backscatter at a resolution of 100 m and 200 m at the agricultural and heathland site.
A good linear correlation between radar backscatter and soil moisture was observed in this study, which is the foundation of the downscaling algorithms. However, this good correlation may be caused by seasonal vegetation variations as indicated in Line 427-429. Please do more analyses to prove that the good correlation between radar backscatter and soil moisture was not induced by vegetation changes.

3. Page 15 Table 5: This table lists eight types of $\beta$ and $\Gamma$ combinations. However, the reviewer cannot follow how the $\beta$ and $\Gamma$ were estimated and the differences between different experiments. Please make more explanations.

4. Page 15 Table 4: This study estimated cluster dependent parameters $\beta$ and $\Gamma$. The parameter of $\beta$ was obtained from linear regression of soil moisture $\theta_{\text{coarse}}$ at coarse resolution and averaged backscatter within this coarse pixel. However, the soil moisture $\theta_{\text{coarse}}$ represents the average soil moisture condition. How can the $\theta_{\text{coarse}}$ be related to backscatters of different land cover types? Please clarify it and make more explanations.

Other comments:
1. Table 1 and Figure 3 can be merged, with $R^2$, bias and RMSE putting in the scatter plots.

2. Page 5 Line 210-211: The Ahlergaarde catchment is covered by 21 SMAP pixels. Please indicate the 21 SMAP pixels in Figure 1 with grids. Are the SMAP pixels in resolution of 9 km by 9 km or 36 km by 36 km?

3. Page 6 Line 230-233: For a deeper investigation of the spatial pattern information content of the Sentinel-1 data, an unsupervised data driven k-means cluster analysis is performed based on four parameters, the mean and the standard deviation of both the VV and the VH backscatter.
How were the mean and the standard deviation values calculated, over temporal variations or spatial variations of radar backscatter? Please clarify.

4. Page 11 Line 375: Heath in Table 2 should be Heathland.

---

## Author Comment (AC1)

Exploring the combined use of SMAP and Sentinel-1 data for downscaling soil moisture beyond the 1 km scale

The authors test the possibility of downscaling SMAP coarse soil moisture to the sub-kilometer resolution using Sentinel-1 SAR data. This paper is interesting and the topic is suitable to HESS. However, I have several major comments the authors should seriously consider.

Dear reviewer, we thank you for thoroughly reading our manuscript and your helpful comments that we believe considerably improved the quality of the revised version of the manuscript. We have added our replies to the reviewer comments and suggestions in *italic* below.

Major comments:

1. This paper directly disaggregates the SMAP coarse soil moisture at 9 km to high resolution using Sentinel-1 SAR backscattering coefficients. It should be noted that the method tested in this paper is based on the assumption of a near-linear relationship between radar backscatter σ0 pp and soil moisture θ at different scales. In order to estimate the parameter β, a time regression is performed under the assumption that the soil roughness and vegetation conditions do not change greatly over a specified temporal window. Meanwhile, the parameter β is NOT invariant in time and space and it depends on vegetation cover and type as well as surface roughness. Therefore, a moving window of β estimation should be adopted when applying this downscaling algorithm to a long time period and the length of time window should be carefully determined. In this study, about 377 images of synchronized SMAP and Sentinel-1 were obtained during the period of January 1, 2017 to May 31, 2019. However, this paper did not describe how to determine the parameter β. In Page 15 Line 445, a temporal window of 40 data points was used to derive seasonal Γ. To derive β?

***Reply:*** *We thank the reviewer for the valid comment. We do agree that β is also influenced by soil roughness and vegetation conditions and hence varies in principal in space and time. One of our major objectives is to test whether the downscaling algorithm can be improved by introducing spatial varying (land use cover dependent) downscaling parameters (β and Γ). In order to take the spatial variability into account we performed the cluster analysis and used this to estimate a land cover dependent β value. The cluster analysis is based on the temporal variation in mean and std. of Sentinel backscatter (both VV and VH).*

*We decided to apply a time invariant (constant) β because the vegetation and surface roughness barely change in the heathland and evergreen pine forest. Of course there is a change in vegetation and roughness in land cover class of agriculture due to land management. However, the crop yield is relatively constant and changes in biomass relatively small which has been studied by* Andreasen et al. (2020)*. Moreover, time series of VV and VH (e.g. Figure 5) show the opposite trend as would be expected if the backscatter signal would be dominated by vegetation and surface roughness. We would expect that a higher amount*

*of vegetation would enhance the volumetric backscatter, both for co- and cross polarization (e.g. Rosenqvist, 2018). However, what we observe is a reduction of the backscatter signal in the growing and peak vegetation periods (spring and summer). On the other hand, in these periods soil moisture is low due to relatively high temperature and evapotranspiration. This trend is mostly observed in the land cover class agriculture, while less visible in the heathland and almost not significant in the forest. Therefore, we believe that applying a time invariant, but spatially varying β is a valid assumption for our study area, representing a classical Danish rural setting.*

*Inspired by the reviewer's comment and also by the first comment of reviewer 2 we performed a seasonal β estimation in order to evaluate if a time varying β would be essential to consider. The following figure shows (a) β estimated over an interval of 3 month\* (Dec-Feb, Mar-May, Jun-Aug, Sep-Nov, representing the seasons in Denmark) and (b) the respective R.[2]*

*\* except for the first interval, starting with January (2 month interval).*

[Figure]

Blue: All data    Red: Agriculture cluster    Green: Heathland cluster

*What we can conclude from this analysis is that there might be a slight seasonality in β with low values in winter and higher values in summer (figure a ). However, if we only consider the β-value with acceptable $R^2$ of above 0.5 (figure b), this trend might not be significant. To estimate a robust time variant β, a dynamic in the range throughout the year would be needed, but what we see is a relatively constant β value, except for the summer. Hence, it is rather difficult to achieve a good estimation of correlation when there is only little variation in the data (during the rest of the year).  On the other hand, we observe that particularly the β estimates for the agriculture cluster (red) deviate significantly from the other data (all=blue and heathland=green). This supports our approach in estimating spatial varying (land cover dependent) but time invariant β.*

*We agree with the reviewer that both the incidence angle correction and β estimation add to the uncertainty of the downscaled product. We acknowledge these comments and will expand the discussion in the manuscript to emphasize these aspects.*

2. Page 14 Line 425-426: The soil moisture derived by CRNS shows a good linear correlation with Sentinel-1 VV and VH backscatter at a resolution of 100 m and 200 m at the agricultural and heathland site. A good linear correlation between radar backscatter and soil moisture was observed in this study, which is the foundation of the downscaling algorithms. However, this good correlation may be caused by seasonal vegetation variations as indicated in Line 427-429. Please do more analyses to prove that the good correlation between radar backscatter and soil moisture was not induced by vegetation changes.

**Reply:** *We understand the concern of the reviewer about the impact of vegetation on the correlation between CRNS and Sentinel backscatter. We believe that the good correlation between CRNS and Sentinel backscatter is soil moisture dominated at the heathland and agricultural site because:*

- *There is little correlation between CRNS and backscatter in the forest where we believe that that the Sentinel (C-band) backscatter is dominated by volume scattering of the pine trees and does barely penetrate to the soil.*
- *There is a high correlation in the heathland where the low vegetation is relatively constant over the seasons. Hence, temporal changes in backscatter are due to soil moisture, which is supported by the good correlation to CRNS signal.*
- *We can observe a good correlation also in the agriculture even though the biomass and vegetation cover changes, as a result of land management. We believe that changes in backscatter are dominantly driven by soil moisture because we observe lower backscatter values in spring/summer during the growing and peak. If the backscatter would be highly influenced by these vegetation changes, we would expect a positive correlation between backscatter and vegetation (higher backscatter signals coinciding with higher vegetation). However, we observe the opposite. Therefore we believe that the backscatter value in the agriculture is mostly influenced by soil moisture.*
- *Previous studies* (Andreasen et al., 2020) *about the CRNS method in the same area showed that the amount of biomass (8.42t/ha) and its seasonal change in the agriculture are rather small and hence the CRNS signal is mostly representative for soil moisture in this area. The reason for this is that the amount of water in the crop is small compared to the amount of water stored in the rootzone.*

*Taking all these points into account, we believe that the strong correlation of backscatter and CRNS is due to soil moisture changes and only insignificantly influenced by vegetation. We will add a condensed version of this line of arguments to the manuscript.*

3. Page 15 Table 5: This table lists eight types of β and Γ combinations. However, the reviewer

cannot follow how the β and Γ were estimated and the differences between different experiments. Please make more explanations.

**Reply:** *Thank you for the comment. We will try to clarify it further below and we will add this more detailed explanation to the manuscript in the Supplemental Material.*

*To investigate whether the downscaling product can be improved by introducing land use cover dependent downscaling parameters (β and Γ), we performed these eight different downscaling tests. Hereby we combined either:*

- *one constant value for β (space and time invariant), estimated as: $\theta_{SMAP}$/VV*
- *or three constant values for β (time invariant), one for each land use cover, estimated as: $\theta_{SMAP}$/$VV_{Cluster}$  (compare Table 4 and Supplemental Material Fig. S3)*

    *with*

- *one constant value for Γ (space and time invariant), estimated as: $\delta VV_{mean}$/$\delta VH_{mean}$*
- *or three constant values for Γ (time invariant), one for each land use cover, estimated as: $\delta VV_{Cluster\_mean}$/$\delta VH_{Cluster\_mean}$*
- *or one time-varying Γ (space invariant), estimated as: $\delta VV_{mean}$/$\delta VH_{mean}$ applying a moving window of 40*
- *or three time-varying Γ, one for each land use cover, estimated as $\delta VV_{Cluster\_mean}$/$\delta VH_{Cluster\_mean}$ applying a moving window of 40*

4. Page 15 Table 4: This study estimated cluster dependent parameters β and Γ. The parameter of β was obtained from linear regression of soil moisture θcoarse at coarse resolution and averaged backscatter within this coarse pixel. However, the soil moisture θcoarse represents the average soil moisture condition. How can the θcoarse be related to backscatters of different land cover types? Please clarify it and make more explanations.

**Reply:** *We follow the reviewer's comment and will explain further how we derive the spatially varying β. Commonly, β relates to the sensitivity of soil moisture to co-polarization radar backscatter ($\sigma_{VV}$) and can be estimated as the slope of a linear regression of $\frac{\theta_{SMAP}}{VV_{coarse}}$ time series. For the land cover dependent β we used the mean and std. of VV and VH time series at different resolutions, e.g. 100 m resolution, in a k-means clustering and derived three clusters at the specific (e.g. 100 m) scale spatially distributed over the entire study area. These three clusters represent the three dominant land use/cover types (heathland, agriculture and forest). The cluster dependent β was consequently estimated based on the linear regression of the time series $\frac{\theta_{SMAP}}{VV_{cluster\ mean}}$. For example for cluster 1, β  was estimated based on the time series of spatial mean of the backscatter signal (VV) of all Sentinel pixels in cluster 1 within the*

*corresponding SMAP pixel* $\frac{\theta_{SMAP}}{VV_{cluster1\ mean}}$. *The distribution of the clusters and the mean and std. of the backscatter signal are shown in figure 4 in the manuscript.*

*We will add this explanation is a condensed version to the manuscript to clarify how β was derived.*

Other comments:

1. Table 1 and Figure 3 can be merged, with R2, bias and RMSE putting in the scatter plots.

**Reply:** *Thank you for this suggestion. We will combine Table 1 and Figure 3.*

2. Page 5 Line 210-211: The Ahlergaarde catchment is covered by 21 SMAP pixels. Please indicate the 21 SMAP pixels in Figure 1 with grids. Are the SMAP pixels in resolution of 9 km by 9 km or 36 km by 36 km?

**Reply:** *We thank the reviewer and understand his/her wish. Since we are using mainly the average of the 21 pixels, we would not like to include the grid in the main manuscript. If the reviewer and editor think it would be useful we would of course add a figure to the supplemental material showing the grid of the SMAP coverage at the study area. The resolution of the 21 SMAP pixel used in our study is the 9km EASE-grid. We will add a sentence in the manuscript to make it more clear.*

3. Page 6 Line 230-233: For a deeper investigation of the spatial pattern information content of the Sentinel-1 data, an unsupervised data driven k-means cluster analysis is performed based on four parameters, the mean and the standard deviation of both the VV and the VH backscatter. How were the mean and the standard deviation values calculated, over temporal variations or spatial variations of radar backscatter? Please clarify.

**Reply:** *We thank the reviewer for the comment and explain further how the cluster analysis was performed. The mean and std. were calculated over temporal variation of the Sentinel (VV and VH) backscatter, which were used for the clustering that resulted in three clusters that are associated with the different land use types as illustrated in figures 4, 5 and 6. We will clarify this aspect in the manuscript.*

4. Page 11 Line 375: Heath in Table 2 should be Heathland.

**Reply:** *Thank you for the comment, we will change as suggested.*

References:

Andreasen, M., Jensen, K. H., Bogena, H., Desilets, D., Zreda, M., & Looms, M. C. (2020).

Cosmic Ray Neutron Soil Moisture Estimation Using Physically Based Site-Specific Conversion Functions. *Water Resources Research*, *56*(11), 1–20. https://doi.org/10.1029/2019WR026588

Rosenqvist, A. (2018). A Layman's Interpretation Guide to L-band and C-band Synthetic Aperture Radar data. In *CEOS* (Issue 2). http://ceos.org

---

## Author Comment (AC2)

The manuscript by Meyer et al., 2021 presented an interesting study in estimating high-resolution soil moisture via the combination of SMAP L3 soil moisture and Sentinel 1 backscatter data. I recommend that the authors address the following comments before considering the paper for publication.

Dear reviewer we thank you for your profound reading and the constructive comments that we believe considerably improved the quality of the revised version of the manuscript. We have added our replies to the reviewer comments and suggestions in *italic* below.

1. Soil moisture at sub-kilometre is indeed in high demand by many regional and local applications. Combination of radiometer and SAR data is definitely valuable and provides a promising way to improve spatial resolution. One major concern is the strong combined effects of incidence angle, biomass and surface roughness on the backscatter. The studies applied a simple methods to calibrate the incidence angle and made a key assumption that ð⌐›½ is invariant in time and space. What impacts of such assumption can influence on the downscaled soil moisture? Furthermore, is that possible to conduct a sensitivity analysis to investigate such impacts?

*Reply: We thank the reviewer for his/her valid comment and understand his/her concerns about the incidence angle correction and assumption used for β estimation.*

*For the incidence angle correction we followed the standard approach that is discussed in* Mladenova et al. (2013) *and applied for a similar purpose in* He et al. (2018): $\sigma_{ref}^0 = \frac{\sigma_{\theta_i}^0 \cos^n(\theta_{ref})}{\cos^n(\theta_i)}$. *Hereby the exponent n is roughness dependent and varies between 0.2 and 3.4* (Mladenova et al., 2013). He et al., (2018) *evaluated a value of n=2 as suitable for a similar application as in this study. In our study area, the mean incidence angle of Sentinel-1 is 36.87° (min 30.25° and max 41.74°). Early on in our study we compared VV and VH time series with and without incidence angle correction, using an exponent of 2. We could see that the correction has a higher effect on VV than on VH. We evaluated that a correction with the exponent of 2 is feasible because it improves the time series by reducing the noise but still showing a dynamic behavior. We do acknowledge that this choice might introduce uncertainty and that applying different exponents might have even further improve our results. However, an extended analysis of the impact of incidence angle correction was out of the scope of our study.*

*As we understand the concern of the reviewer about the β estimation is similar to comment 1 raised by reviewer 1, we hope that we already there answered satisfactorily. For an easier reading we add our reply to reviewer 1 here again:*

*"One of our major objectives is to test whether the downscaling algorithm can be improved by introducing spatial varying (land use cover dependent) downscaling parameters (β and Γ). In order to take the spatial variability into account we performed the cluster analysis and used this to estimate a land cover dependent β. The cluster analysis is based on the temporal variation in mean and std. of Sentinel backscatter (both VV and VH).*

We decided to apply a time invariant (constant) β because the vegetation and surface roughness barely change in the heathland and evergreen pine forest. Of course in the agriculture due to land management there is a change in vegetation and roughness, however, the crop yield is relatively constant and changes in biomass relatively small which has been studied by Andreasen et al. (2020). Moreover, time series of VV and VH (e.g. Figure 5 ) show the opposite trend as would be expected if the backscatter signal would be dominated by vegetation and surface roughness. We would expect that a higher amount of vegetation would enhanced the volumetric backscatter, both in co- and cross polarization (e.g. Rosenqvist, 2018). But, what we observe is a reduction of the backscatter signal in the growing and peak vegetation periods (spring and summer). On the other hand in these periods soil moisture is low due to relatively high temperature and evapotranspiration. This trend is mostly observed in the agriculture while less visible in the heathland and almost not significant in the forest. Therefore, we believe that applying a time invariant, but spatial varying β is a valid assumption for our study system.

Inspired by the reviewer's comment and also by the first comment of reviewer 2 we performed a seasonal β estimation in order to evaluate if a time varying β would be essential to consider. The following figure shows (a) β estimated over an interval of 3 month* (Dec-Feb, Mar-May, Jun-Aug, Sep-Nov, representing the seasons in Denmark) and (b) the respective R.$^2$

* except for the first interval, starting with January (2 month interval).

[Figure]

Blue: All data    Red: Agriculture cluster    Green: Heathland cluster

What we can conclude from this analysis is that there might be a slight seasonality in β with low values in winter and higher values in summer (figure ,a ). However, if we only consider the β -value with acceptable $R^2$ of above 0.5 (figure , b), this trend might not be significant. On the other hand, we observe that particularly the β estimates for the agriculture cluster (red) deviate significantly from the other data (all=blue and heathland=green). This supports our approach in estimating spatial varying (land cover dependent) but time invariant β."

*The reviewer is right that both the incidence angle correction and β estimation add to the uncertainty of the downscaled product. We will emphasize these aspects in the discussion of our manuscript.*

2. CRNS data was used as a reference to evaluate satellite-based soil moisture. Since CRNS neutron is also influenced by vegetation water content, did you calibrate such impacts in deriving volumetric soil moisture? The CRNS also has variable spatial and vertical footprints. Not sure if the direct comparison with satellite surface soil moisture is appropriate. Is that possible to consider such representative errors in your evaluation?

**Reply:** *We thank the reviewer for his/her valid comment and will explain the CRNS data set in detail. A short version of the explanation will be added to the manuscript.*

*The influence of the vegetation water content on the CRNS estimated soil moisture is low. In the Heathland and pine forest, there is very limited change in vegetation cover. At the agricultural site the amount of biomass is relatively low (8 t/ha consisting of ~15% cellulose and 85% water, Andreasen et al., 2020). Andreasen et al. (2020) also tested the impact of the vegetation cover on the CRN intensity using field measurements of neutrons at two energy ranges and neutron transport modeling (Monte Carlo N-Particle code version 6, MCNP6) of the agricultural field site. Their analysis showed very little impact of the vegetation cover compared to bare soil conditions (Andreasen et al., 2020, Figure 4).*

*The second concern of the reviewer relates to the variability in spatial and vertical footprints. We do agree with the reviewer that the CRNS footprint varies in space and time. However, it should be noted that the sensor sensitivity is highest in the close vicinity of the probe (86% within a radius of 200m) and decreases exponentially with distance from the sensor. The CRNS sensors were installed at three different land use/cover types in 2013/2014 and data collection is still ongoing. The location of the sensors has been carefully chosen. They are set up in a way that they are in the same soil type and placed far enough from the next land use/cover type to prevent influence/mixture of different LUC signals. Furthermore, Ahlergaarde catchment is situated on a glacial outwash plain, and the study area is characterized by homogeneous soil (sandy and stratified soil with similar chemical composition). Therefore, we do not expect changes in the vertical and horizontal footprint area to affect the CRN signal significantly. A network of capacitance probes (please note that this network is not the same as used in our manuscript, but specifically set up to validate/compare the CRNS estimates), TDR measurements and soil probes are placed/taken strategically in the vicinity of the CRNS sensors. The long time series of CRNS estimated soil moisture has been shown to be very robust in comparison to the average of these measurements* (Andreasen et al., 2020). *The same data set has been successfully used to improve the closing of the water balance by* Denager et al. (2020)*.*

*The CRNS estimates of soil moisture are subject to uncertainties, but we believe that at this stage and for our purpose the method is better than any other available technology, and particularly because the spatial scale is similar to the envisaged downscaled soil moisture product.*

3. Another comment is regarding the validation of your downscaled soil moisture. As authors described, small modifications to the downscaling approach can induce significant changes in spatial patterns, it is therefore challenging to identify the best approach. I agree with such statement, but also want to ask how to distinguish noise and real soil moisture patterns? Direct comparison with CRNS might not sufficient due to the scale mismatch and high diversity of soil properties. In addition, can you give some practical advice or outlook on how to generate sub-kilometre soil moisture products, which can be used for fine-scale applications?

*Reply: Thank you for the comment. We understand the concern of the reviewer and try to explain our approach. We believe that the application of CRNS soil moisture estimates, in addition to more conventional capacitance probes measurement constitute a major improvement for validation of remotely sensed and downscaled soil moisture products. The biggest advantage is the similar scale of horizontal sensitivity of the downscaled product and the CRNS which is in the order of few hundreds meter. To minimize the impact of noise we smoothed the Sentinel backscatter, spatially (a minimum aggregation to 20 m) and temporally, by applying a moving average of five images. Moreover, meaningful trends in the time series are visible that follow the season and the expected soil moisture. By validating against different independent ground measurements (CRNS and capacitance probes), each of them of course has advantages and disadvantages, we try to enhance the reliability of our approach and results. One of our main challenge was to get a reliable data set for validation. In our study area the capacitance probe time series had many gaps. Moreover, we think that the CRNS does not have as bad a mismatch in scale as other soil moisture products, e.g. capacitance probes.*

*We fully agree with the reviewer that one of the major challenge in current soil moisture research is the differentiation between noise and soil moisture and the different scales of downscaled product and validation data. We hope that the application of new technologies, e.g. roving CRNS, can address these issues in the future.*

*Our advice for future research in this area includes getting a high quality dataset in the same scale for validating of the downscaled product. This could be achieved e.g. by enlarging the study area so that more CRNS stations can be used (e.g. it exist a network of about 50 CRNS stations across Europe). The use of such a network for calibration of the downscaling parameters β and Γ might be successful and hence improve the algorithm substantially.*

4. In cluster analysis, 20m, 100m, 1000m were selected and analysed. What is the criterial to choose these scales? For the downscaled soil moisture, 100 m is presented as "the downscaled sub-kilometre" product. Does it mean it is the tradeoff between quality and resolution?

*Reply: We thank the reviewer for this remark. Actually, we performed the whole analysis with many different resolutions between 20 m and 1000 m. The objective is to investigate the sub kilometer scale because previous studies aimed for the 1km scale. For the perspective in applying the downscaled soil moisture for catchment hydrological questions, a resolution in the hundreds meter is favorable. Moreover, the CRNS footprint lies in the 100-200 m resolution. In order to be concise in our results we choose to mainly show the 100 m results in detail and occasionally show also the 20 m and 1000 m results, but not the 50 m, 200 m, 400*

*m, 500 m and 800 m which we had also performed. Hence, the 100 m resolution is not a trade of, as the 200 m may look similarly meaningful. However, from our results we can clearly see, that at the 1000 m resolution a lot of valuable information is lost, e.g. compare figures 4, 6 and 8. The choice of showing the 100 m resolution results from the purpose (future application in hydrological modelling), available ground data (CRNS with a footprint of 100-200 m) and aiming for conciseness.*

5. Remote the comma in the title.

**Reply:** *Thank you for the comment. We will change as suggested.*

6. Caption figure 4: c is backscatter and d is db.

**Reply:** *Thank you for the comment. We will double check to be sure there is no mistake.*

References:

Andreasen, M., Jensen, K. H., Bogena, H., Desilets, D., Zreda, M., & Looms, M. C. (2020). Cosmic Ray Neutron Soil Moisture Estimation Using Physically Based Site-Specific Conversion Functions. *Water Resources Research*, *56*(11), 1–20. https://doi.org/10.1029/2019WR026588

Denager, T., Looms, M. C., Sonnenborg, T. O., & Jensen, K. H. (2020). Comparison of evapotranspiration estimates using the water balance and the eddy covariance methods. *Vadose Zone Journal*, *19*(1), 1–21. https://doi.org/10.1002/vzj2.20032

He, L., Hong, Y., Wu, X., Ye, N., Walker, J. P., & Chen, X. (2018). Investigation of SMAP Active – Passive Downscaling Algorithms Using Combined Sentinel-1 SAR and SMAP Radiometer Data. *IEEE Transactions on Geoscience and Remote Sensing*, *56*(8), 4906–4918. https://doi.org/10.1109/TGRS.2018.2842153

Mladenova, I. E., Jackson, T. J., Bindlish, R., Member, S., Hensley, S., & Member, S. (2013). Incidence Angle Normalization of Radar Backscatter Data. *IEEE Transactions on Geoscience and Remote Sensing*, *51*(2), 1791–1804. https://doi.org/10.1109/TGRS.2012.2205264

Rosenqvist, A. (2018). A Layman's Interpretation Guide to L-band and C-band Synthetic Aperture Radar data. In *CEOS* (Issue 2). http://ceos.org

---

## Author Response (AR2)

Dear editor and reviewers,

We thank you for the thorough handling of our manuscript and are happy that it is accepted for publication.

We have carefully considered the final comments in the author group and have decided not to include additional references or elaborate on the limitations and scale mismatch of soil moisture retrievals from satellites and cosmic ray neutron sensor (CRNS). Below we present our reasoning.

We have included references to existing literature concerning the comparison of satellite derived soil moisture and CRNS derived soil moisture at their original measurement scale (e.g. line 112). Here we discuss the scale mismatch. The objective of our study is to evaluate downscaled satellite derived soil moisture at the same scale as the measurement scale of the CRNS. The advantage being the similar scale and hence no significant scale mismatch.

In order not to distract the reader from this main point we do not want to elaborate further on the general and well-studied aspects of scale mismatch between non-downscaled satellite and ground measurements. Furthermore, we are not aware of which specific papers the reviewer has in mind.

Finally, we agree that the COSMOS-Europe paper is very valuable for future downscaling exercises. That is why we have already included it in our manuscript.

Sincerely,

Rena Meyer (in the name of all authors)